# R-spondin 3 deletion induces Erk phosphorylation to enhance Wnt signaling and promote bone formation in the appendicular skeleton

Kenichi Nagano[1], Kei Yamana[1], Hiroaki Saito[1], Riku Kiviranta[1], Ana Clara Pedroni[1], Dhairya Raval[1], Christof Niehrs[2,3], Francesca Gori[1]*, Roland Baron[1,4,5]*

[1]School of Dental Medicine, Harvard University, Boston, United States; [2]German Cancer Research Center, DKFZ-ZMBH Alliance, Heidelberg, Germany; [3]Institute of Molecular Biology (IMB), Mainz, Germany; [4]Department of Medicine, Harvard Medical School, Boston, United States; [5]Endocrine Unit, Massachusetts General Hospital, Boston, United States

*For correspondence:
francesca_gori@hsdm.harvard.edu (FG);
Roland_baron@hsdm.harvard.edu (RB)

**Competing interest:** The authors declare that no competing interests exist.

**Abstract** Activation of Wnt signaling leads to high bone density. The R-spondin family of four secreted glycoproteins (Rspo1-4) amplifies Wnt signaling. In humans, RSPO3 variants are strongly associated with bone density. Here, we investigated the role of Rspo3 in skeletal homeostasis in mice. Using a comprehensive set of mouse genetic and mechanistic studies, we show that in the appendicular skeleton, *Rspo3* haplo-insufficiency and *Rspo3* targeted deletion in *Runx2*+ osteo-progenitors lead to an increase in trabecular bone mass, with increased number of osteoblasts and bone formation. In contrast and highlighting the complexity of Wnt signaling in the regulation of skeletal homeostasis, we show that *Rspo3* deletion in osteoprogenitors results in the opposite phenotype in the axial skeleton, i.e., low vertebral trabecular bone mass. Mechanistically, *Rspo3* deficiency impairs the inhibitory effect of Dkk1 on Wnt signaling activation and bone mass. We demonstrate that *Rspo3* deficiency leads to activation of Erk signaling which in turn, stabilizes β-catenin and Wnt signaling activation. Our data demonstrate that *Rspo3* haplo-insufficiency/deficiency boosts canonical Wnt signaling by activating Erk signaling, to favor osteoblastogenesis, bone formation, and bone mass.

## Editor's evaluation

This seminal paper describes the divergent effects of Rspo3 haploinsufficiency on appendicular versus the axial skeletal in mice and, in doing so, highlights the differential regulation of the Wnt signaling pathway in a tissue-specific manner. Thus, while Rspo3 deficiency in osteoprogenitor cells increases bone mass in the appendicular skeleton, it causes osteopenia of vertebral bone. The study therefore not only identifies RSPO3, variants of which are associated with bone density in people, as a target for enhancing bone density in fracture-prone appendicular sites in the aging population, but also brings forth caution in the interpretation of single-site studies of the skeleton in most general terms.

## Introduction

The Wnt signaling pathway controls cell fate decisions and tissue homeostasis during development and in the adult (*Clevers and Nusse, 2012*; *Nusse and Clevers, 2017*). It is also involved in skeletal

development and essential for the regulation of bone mass in the adult skeleton (*Baron and Kneissel, 2013*; *Gori and Baron, 2021*; *Huybrechts et al., 2020*). Genetic or therapeutic activation of canonical Wnt signaling is associated with increased bone mass (*Baron and Kneissel, 2013*) and current therapeutic approaches aim at activating this pathway in patients with osteoporosis or osteogenesis imperfecta for instance (*Baron et al., 2020*).

Wnt signaling involves a large number of receptors and co-receptors, and of endogenous agonists and antagonists that, together, tightly regulate its activation (*Clevers and Nusse, 2012*; *Nusse and Clevers, 2017*; *Baron and Kneissel, 2013*; *Gori and Baron, 2021*). Due to this complexity, and even though Wnt signaling has been studied extensively in recent years in bone, several aspects of the mechanisms by which it regulates bone mass remain unclear. Similarly, the specific downstream events regulated by the various components of the Wnt signaling machinery and their interaction with other signaling cascades remain puzzling.

In this context, the fact that several studies have demonstrated that the four Roof plate-specific spondin, R-spondins (Rspo1 to 4), synergize with Wnt ligands to activate Wnt signaling (*Gori and Baron, 2021*; *de Lau et al., 2014*; *de Lau et al., 2012*; *Knight and Hankenson, 2014*; *Lu et al., 2008*; *Nagano, 2019*; *Raslan and Yoon, 2019*; *Shi et al., 2016*) raises the question of their potential role in skeletal development and homeostasis. This enhancement of Wnt activity is attributed to the ability of Rspos to prevent the ubiquitination and degradation of the Lrp5/6/Fzd receptor complex via Lgr4-6, closely related orphans of the leucin-rich repeat containing G protein-coupled receptors, and the transmembrane E3 ubiquitin ligases ring finger 43 (Rnf43) and zinc and ring finger 3 (Znrf3), sensitizing cells to Wnt ligands (*de Lau et al., 2014*; *Glinka et al., 2011*; *Hao et al., 2012*; *Ruffner et al., 2012*; *Wang et al., 2013*; *Xie et al., 2013*). Although the role of Lgr receptors in the effects of Rspos is well established, recent reports have shown that Rspo2 and Rspo3 can also enhance Wnt signaling independently of Lgr receptors, possibly by acting as direct antagonist ligands to RNF43 and ZNRF3 (*Nagano, 2019*; *Lebensohn and Rohatgi, 2018*; *Szenker-Ravi et al., 2018*).

Of particular interest is the fact that, in contrast to the many studies that have reported that Rspos co-activate Wnt signaling, studies in Zebrafish have indicated that Rspo3 can also function as a negative regulator of canonical Wnt signaling during dorsoventral and anteroposterior patterning (*Rong et al., 2014*; *Wu et al., 2014*). Additionally, it has been shown that Rspos can potentiate non-canonical Wnt cascades, such as the Wnt/PCP signaling (*Glinka et al., 2011*; *Hao et al., 2012*) and can function as antagonists of BMPR1 in *Xenopus* (*Lee et al., 2020*), two events that could have a negative impact on bone formation and bone mass. Thus, the mechanisms involved in the Rspos/Wnt signaling axis and their influence on skeletal homeostasis appear to be more complex in vivo.

The four Rspos belong to a family of cysteine-rich secreted glycoproteins with high structural similarity and 60% sequence homology (*de Lau et al., 2012*; *Nagano, 2019*). Although all Rspos are expressed in bone during development, they have specific and unique functions as reflected by the findings that their deletion leads to different phenotypes (*Lu et al., 2008*; *Shi et al., 2016*; *Aoki et al., 2007*; *Neufeld et al., 2012*; *Ishii et al., 2008*; *Kazanskaya et al., 2008*; *Knight et al., 2018*; *Park et al., 2018*; *Parma et al., 2006*; *Wei et al., 2007*). Within the Rspo family, Rspo3 is of particular interest for bone because it is highly expressed in skeletal elements during mouse development (*Nam et al., 2007*; *Alhazmi et al., 2021*) and several human GWAS studies have shown that RSPO3 SNPs are strongly associated with bone mineral density and risk of fracture (*Duncan et al., 2011*; *Estrada et al., 2012*; *Lee et al., 2010*; *Medina-Gomez et al., 2012*; *Richards et al., 2008*; *Richards et al., 2012*; *Nilsson et al., 2021*). Not surprisingly, in vitro studies have shown that overexpression of, or treatment with, Rspos can enhance Wnt-ligand mediated osteoblast (OB) differentiation (*Lu et al., 2008*; *Knight et al., 2018*; *Zhu et al., 2016*). However, similar to the Zebrafish and *Xenopus* studies mentioned above, it has also been reported that Rspo3 may function as a negative regulator of osteoblast differentiation in vitro (*Zhang et al., 2017*). Thus, taking into account the entire literature raises questions about the true net influence of Rspos, and in particular Rspo3, on skeletal homeostasis. We have recently reported that *Rspo3* is expressed in osteoprogenitors in the craniofacial complex in both mice and Zebrafish (*Alhazmi et al., 2021*). However, Rspo3 disruption in Zebrafish has only mild effects on larval craniofacial cartilage skeleton (*Alhazmi et al., 2021*) and in mice its haplo-insufficiency does not appear to affect craniofacial development. Nilsson et al. have recently reported that targeted deletion of *Rspo3* in osteoblast precursors in mice leads to low bone mass in the axial skeleton (L5) at

12 weeks of age, they examined only the axial skeleton where no changes in cellular and dynamic parameters were observed (*Nilsson et al., 2021*).

In the current studies, we have further investigated the role of Rspo3 in skeletal homeostasis. We found that if, as recently reported (*Nilsson et al., 2021*; *Nilsson et al., 2022*), targeted deletion of *Rspo3* in Runx2$^+$ cells decreases bone mass in the axial skeleton, both global *Rspo3* haplo-insufficiency and targeted deletion of *Rspo3* in Runx2$^+$ osteoblast precursors lead to a marked increase in trabecular bone mass in the appendicular skeleton in both male and female mice, mainly as a result of increased bone formation. Mechanistically, we found that Rspo3 deletion leads to increased Erk phosphorylation, and, similar to increased Rspo3, stabilization of β-catenin and, activation of Wnt signaling, revealing a novel Rspo3/Erk/Wnt signaling axis that contributes to the regulation of skeletal homeostasis.

## Results

### *Rspo3* haplo-insufficiency increases bone formation and trabecular bone mass

Rspo3 is expressed in bone during mouse skeletal development (*Nam et al., 2007*). We assessed *Rspo3* mRNA levels in primary calvarial osteoblasts (cOBs) and found that it is expressed in these cells and that its expression increases significantly during OB differentiation (*Figure 1—figure supplement 1a*). To determine the physiological role of Rspo3 in skeletal homeostasis, we first used mice in which *Rspo3* has been germline-deleted (*Kazanskaya et al., 2008*). As global deletion of *Rspo3* leads to lethality by E9.5 (*Aoki et al., 2007*; *Kazanskaya et al., 2008*), before skeletal development, we analyzed mice lacking only one *Rspo3* allele (*Rspo3$^{+/-}$*) and their WT littermates. *Rspo3$^{+/-}$* mice are born at the expected Mendelian ratio and their axial and appendicular skeleton develop normally (*Figure 1—figure supplement 1b*). *Rspo3$^{+/-}$* mice continue to be healthy and do not develop any detectable pathology as they age (up to one year) (data not shown). We confirmed that *Rspo3* mRNA levels were significantly decreased (50%) in *Rspo3$^{+/-}$* mice in both marrow-depleted long bones and in cultured cOBs compared with WT mice (*Figure 1a* and *Figure 1—figure supplement 1a*). Contrary to expectations, two-way ANOVA analysis of the skeletal phenotype at 6, 12, and 18 wk of age (*Figure 1b*, *Table 1*, *Figure 1—figure supplement 1c*, *Supplementary file 1*), revealed highly significant ($P<0.001$) anabolic effects of *Rspo3* haplo-insufficiency on structural, cellular, and dynamic parameters in both female and male mice. The skeletal phenotype of *Rspo3$^{+/-}$* male and female mice is overall characterized by an increase in trabecular bone mass with high bone formation, mineral apposition rate and OB number and surface whereas bone resorption parameters are not affected, as indicated by osteoclast (OC) number and surface as well as eroded surface (ES/BS) (*Figure 1b*, *Table 1*, *Supplementary file 1*).

As expected, Two-way ANOVA also demonstrated an effect of age, affecting primarily the structural parameters (*Figure 1b*, *Table 1* and *Supplementary file 1*). At 12 wk of age *Rspo3* haplo-insufficiency led to a significant increase in trabecular bone mass (BV/TV), trabecular thickness (Tb. Th.), and trabecular number (Tb.N) (*Figure 1c–d*, *Table 1* and *Supplementary file 1*). Trabecular bone resorption parameters (Oc.S/B.Pm and N.Oc/B.Pm) were not changed in both *Rspo3$^{+/-}$* male and female mice. In contrast, *Rspo3$^{+/-}$* mice exhibited an increase in trabecular bone formation parameters (BFR/BS) (*Figure 1d–e*, *Table 1* and *Supplementary file 1*). This increase in BFR was associated with an increase in mineral apposition rate (MAR), indicating a marked increase in the activity of individual OBs in *Rspo3$^{+/-}$* mice, in addition to the increase in their numbers (N.Ob/B.pm, *Figure 1d*, *Table 1* and *Supplementary file 1*). Consistent with these results, the osteoid surface (OS/BS) and the OB surface (Ob.S/B.Pm) were also significantly increased in both sexes (*Table 1* and *Supplementary file 1*). Despite these marked effects on trabecular bone, cortical bone parameters were not significantly changed in *Rspo3$^{+/-}$* female and male mice at 12 wk of age (*Figure 1d*, *Supplementary file 2* and *Supplementary file 3*). In contrast to the observed effects in the tibia, no significant differences in bone mass or cellular activities were noted in the vertebrae (L5) at 12 wk of age (*Figure 1—figure supplement 2* and *Supplementary file 4*), indicating that *Rspo3* haplo-insufficiency affects preferentially the trabecular bone compartment in long bones.

Thus, in contrast with the expectation that decreasing the expression of a Wnt signaling potentiator might lead to a decrease in bone mass, our data clearly indicates that in the appendicular skeleton

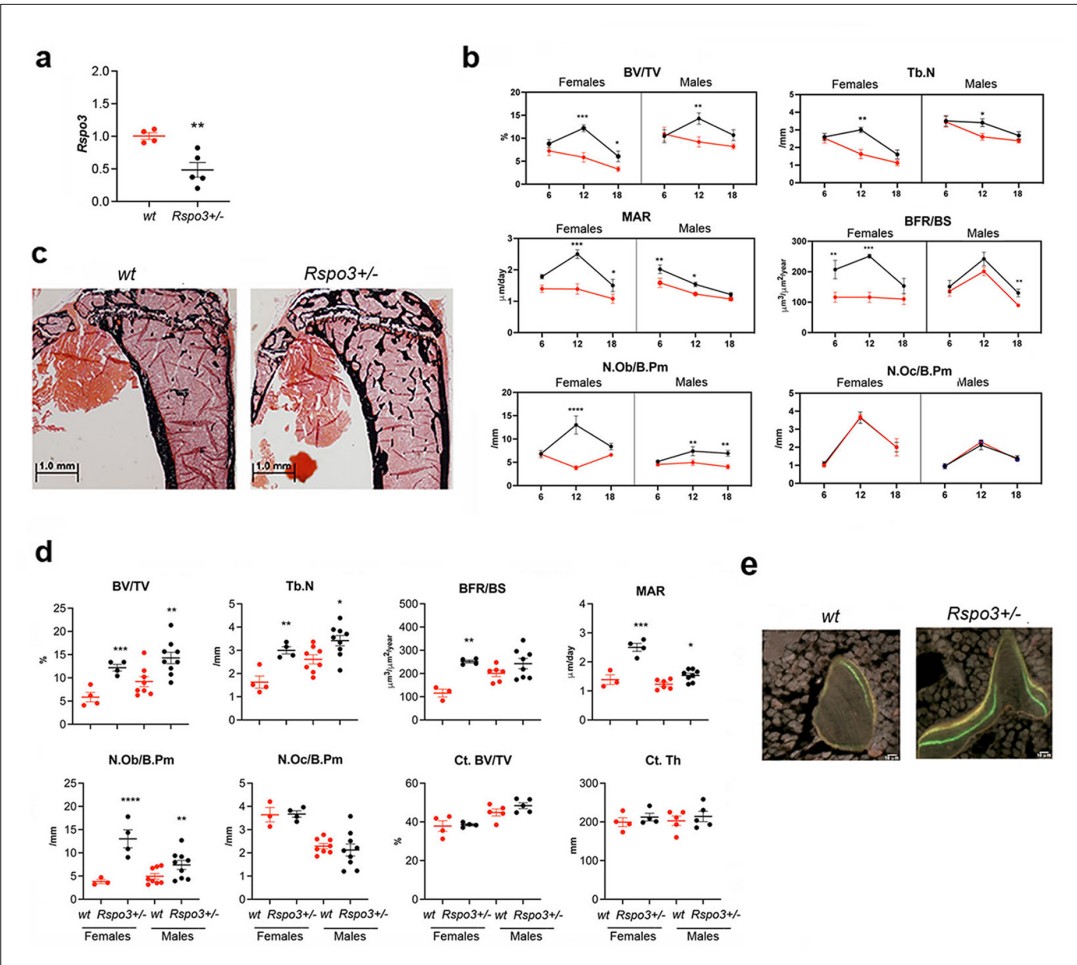

**Figure 1.** Skeletal phenotype of mice with *Rspo3* haplo-insufficiency. (**a**) *Rspo3* expression in marrow depleted long bones, isolated from WT and *Rspo3⁺/⁻* mice (n=4–5). Data show all samples and are the mean ± SEM **p<0.005 by unpaired Student's t-test. (**b**) Histomorphometric analysis in 6, 12, and 18 wk males and females (red line = WT, black line = *Rspo3⁺/⁻*). (n=3–9). Data are the mean ± SEM. Two-way ANOVA followed by Fisher's LSD test. *p<0.05, **p<0.005, ****p<0.0001. (**c**) Representative images of Von Kossa staining in 12 wk-old WT and *Rspo3⁺/⁻* male tibiae. (n=8-9). Scale bars = 1.0 mm. (**d**) Histomorphometric analysis of 12 wk-old WT and *Rspo3⁺/⁻* female and male tibiae (n=3–9). Data show all samples and the mean ± SEM. *=p<0.05, **p<0.005 by unpaired Student's t-test. Red circles = WT, Black circles = *Rspo3⁺/⁻*. (**e**) Representative images of double labeling in trabecular bone in 12 wk-old WT and *Rspo3⁺/⁻* male tibiae. (n=8-9). Scale bars = 10 μm.

The online version of this article includes the following figure supplement(s) for figure 1:

**Figure supplement 1.** Skeletal phenotype of mice with *Rspo3* haploinsufficiency.

**Figure supplement 2.** Axial skeletal phenotype of *Rspo3⁺/⁻* mice.

**Figure supplement 3.** *Rspo3* haploinsufficiency does not affect osteoclastogenesis.

*Rspo3* haplo-insufficiency induces an increase in trabecular bone mass due to a significant increase in bone formation, with no changes in bone resorption in both males and females. In agreement with our in vivo observations, we found that, although *Rspo3* is expressed in bone marrow macrophage (BMM)-derived OCs (*Figure 1—figure supplement 3a*), there were no significant differences in the formation of TRAP⁺ multinucleated cells and in the expression of OC marker genes (*Ctsk*, *Trap*, *Nfatc1*) between BMM cultures from WT and *Rspo3⁺/⁻* in response to M-CSF and RANKL (*Figure 1—figure supplement 3b and c*). In addition, mix-and-matched co-cultures of cOBs and BMMs from WT or *Rspo3⁺/⁻* mice confirmed that *Rspo3* haplo-insufficiency does not affect osteoclastogenesis, whether directly or indirectly (*Figure 1—figure supplement 3d*).

**Table 1.** Histomorphometric analysis of WT and *Rspo3*$^{+/-}$ females.

| Parameters | 6 wk | | 12 wk | | 18 wk | | Two way ANOVA | | |
| --- | --- | --- | --- | --- | --- | --- | --- | --- | --- |
| | WT (n=6) | Rspo3$^{+/-}$ (n=7) | WT (n=4) | Rspo3$^{+/-}$ (n=4) | WT (n=8) | Rspo3$^{+/-}$ (n=6) | Genotype | Age | Interaction |
| BV/TV (%) | 7.24±0.98 | 8.8±0.88 | 5.84±1.01 | 12.2±0.72*** | 3.28±0.51 | 6.06±1.1* | <0.0001 | <0.0001 | NS |
| Tb.Th (µm) | 28.3±0.91 | 33.6±1.5* | 35.8±2.54 | 40.7±1.31 | 28.2±1.55 | 36.8±2.7** | 0.0003 | 0.0032 | NS |
| Tb.N (/mm) | 2.53±0.27 | 2.6±0.20 | 1.63±0.26 | 3.00±0.15** | 1.13±0.16 | 1.61±0.25 | 0.0025 | <0.0001 | 0.0429 |
| Tb.Sp (µm) | 397±52.6 | 368±26.4 | 618±86.8 | 297±17.1* | 1050±192 | 690±133 | 0.0378 | 0.0008 | NS |
| MAR (µm/day) | 1.4±0.11 | 1.8±0.06 | 1.39±0.17 | 2.5±0.14*** | 1.08±0.15 | 1.5±0.20* | <0.0001 | 0.0016 | NS |
| MS/BS (%) | 22.2±2.06 | 31.4±4.05* | 23.0±3.25 | 27.8±1.63 | 26.8±1.90 | 27.3±1.55 | 0.0394 | NS | NS |
| BFR/BS (µm³/µm²/year) | 116.2±16.6 | 207±30.4** | 116±16.9 | 251±6.43** | 110±17.6 | 153±25.1 | <0.0001 | NS | NS |
| N.Ob/B.Pm (/mm) | 6.72±0.79 | 6.85±0.64 | 3.83±0.45 | 13±1.94**** | 6.60±0.39 | 8.4±0.69 | <0.0001 | NS | 0.0001 |
| Ob.S/B.Pm (%) | 10.3±1.20 | 10.5±1.44 | 4.89±0.38 | 16.3±2.2**** | 9.66±0.52 | 12.3±0.53 | <0.0001 | NS | 0.0007 |
| OS/BS (%) | 4.68±0.78 | 5.51±1.23 | 2.51±0.30 | 9.17±0.77*** | 4.84±0.66 | 8.3±0.74** | <0.0001 | NS | 0.0263 |
| O.Th (µm) | 3.89±0.30 | 4.64±0.37 | 2.76±0.13 | 4.36±0.09* | 2.78±0.37 | 4.24±0.3** | 0.0003 | NS | NS |
| N.Oc/B.Pm (/mm) | 1.08±0.13 | 0.99±0.11 | 3.64±0.31 | 3.67±0.14 | 2.01±0.26 | 2±0.48 | NS | <0.0001 | NS |
| Oc.S/B.Pm (%) | 2.98±0.40 | 3.07±0.37 | 7.89±0.42 | 8.77±0.55 | 5.85±0.86 | 5.7±1.17 | NS | <0.0001 | NS |
| ES/BS (%) | 4.41±1.06 | 4.05±0.42 | 1.67±0.48 | 2.96±0.43 | 6.85±0.82 | 6.74±1.28 | NS | <0.0001 | NS |

Data are expressed as Mean ± SEM. Two way ANOVA followed by Fisher's LSD post-hoc test. *p<0.05, **=p < 0.005, ***=p < 0.001, ****=p < 0.0001 compared with age-matched WT females.

## *Rspo3* haplo-insufficiency leads to an increase in bone marrow precursor cells and in their osteoblast potential

Given that the OB number was significantly increased in *Rspo3*[+/-] mice compared with WT mice (*Figure 1*, *Table 1* and *Supplementary file 1*), we determined whether this was associated with an increase in the population of precursor cells. Bone marrow flow cytometry showed that while the total number of bone stromal cells (Lin[-]CD45[-]) was not significantly affected by Rspo3 haplo-insufficiency (4047±1245 in WT compared with 5867+2382 in *Rspo3*[+/-], mean ± SEM n=10), the mesenchymal stromal cells (MSC) population (defined here as Lin[-]CD45[-]CD31[-]CD51[+]Sca-1[+]) (*Schepers et al., 2013*) was significantly increased in *Rspo3*[+/-] mice compared with WT littermates (p=0.015) (*Figure 2a and b*). Consistent with these findings and with the observed increase in OB number and bone formation, Rspo3 haplo-insufficiency significantly increased CFU-F (p=0.04), and CFU-OB (p=0.034) formation (*Figure 2c*). Importantly, while treatment with recombinant Rspo3 did not affect the CFU-F and CFU-OB potential of WT BMSCs, it showed a tendency to decrease CFU-F and rescued significantly, i.e., prevented, the *Rspo3* haplo-insufficiency-dependent increase in CFU-OB formation (*Figure 2c*). These data show that the changes induced by *Rspo3* haplo-insufficiency cell-autonomously affect the bone marrow MSC lineage and induce an increase in the pool of progenitor cells with OB potential.

## Specific deletion of *Rspo3* in Runx2[+] cells leads to high bone mass in the appendicular skeleton and low bone mass in the axial skeleton

To confirm the cell-autonomous effect of Rspo3 to cells of the OB lineage in vivo, we generated mice with deletion of *Rspo3* in the OB lineage (Rspo3-OB-cKO) by crossing *Rspo3*[fl/fl] mice with the OB progenitors-specific *Runx2-Cre* mice (*Rauch et al., 2010*; *Movérare-Skrtic et al., 2014*). OB lineage-targeted deletion of *Rspo3* (*Figure 3a*) reproduced the skeletal phenotype seen in the appendicular skeleton of *Rspo3*[+/-] mice as indicated by a significant increase in BV/TV (females p=0.017, males p<0.0001), MAR (females p=0.0034, males p<0.0001), BFR/BS (females p=0.0012, males p<0.0001) and N.Ob/B.Pm (p=0.033, males p=0.007) in Rspo3-OB-cKO compared with their control (*Rspo3*[fl]) littermates. Once again there were no changes in OC parameters (*Supplementary file 5*). µCT analysis of the femur, confirmed a significant increase in BV/TV, Tb.N, and Conn.D in both males and females (*Figure 3d*).

Very surprisingly, however, and although the basal expression level of *Rspo3* and the efficiency of deletion in long bones and vertebrae were similar (*Figure 3a* and *Figure 4a*), L5 µCT analysis showed a significant decrease in BV/TV, Tb.N, Tb.Th, and Conn.D accompanied by a marked increase in Tb.Sp. and SMI in Rspo3-OB-cKO compared with their control (*Rspo3*[fl]) male and female littermates (*Figure 4b*), confirming a recent report (*Nilsson et al., 2022*). Bone histomorphometry analysis showed that structural parameters were also significantly decreased but only in Rspo3-OB-cKO males compared with the control group (*Figure 4c–d* and *Supplementary file 6*). No significant changes were seen in the dynamic parameters. While the N.Ob/B.Pm and Ob.S/BS were not changed, the N.Oc/ B.Pm was significantly increased in Rspo3-OB-cKO males but not females, compared with the control group (*Figure 4d* and *Supplementary file 6*), providing a possible explanation for the decreased bone mass, at least in males. Thus, our studies show that targeted deletion of *Rspo3* in Runx2[+] osteoprogenitors leads to an opposite bone phenotype in the appendicular and the axial skeleton. Excluding any effect of the transgene or the floxed allele on the differential effect of *Rspo3* targeted deletion in the axial and appendicular skeleton, µCT analysis of the femur and L5 vertebrae did not show any structural difference in BV/TV, Tb.N, Tb.Th, or Tb.Sp between *Runx2-Cre*, WT, and *Rspo3*[fl] mice (*Figure 3* and *Figure 3—figure supplement 1*).

## *Rspo3* haplo-insufficiency and deletion lead to β-catenin stabilization

Independent of the differences, at least at the 12 wk time point, between axial and appendicular skeletal sites, our results showed a clear increase in OB progenitors in the bone marrow of *Rspo3*[+/-] mice (*Figure 2*) and given that trabecular bone is the main compartment affected by *Rspo3* haplo-insufficiency, we then explored how depletion of *Rspo3* was affecting Wnt signaling in BMSCs. Surprisingly, but consistent with our observations on bone formation, *Rspo3* haplo-insufficiency led to a remarkable increase in the expression of known canonical Wnt target genes *Axin2* and *Tcf7* and increased active β-catenin levels (*Figure 5a and b*). In addition, *Runx2* expression was also significantly increased in *Rspo3*[+/-] BMSCs (*Figure 5a*). In addition, *Rspo3* haplo-insufficiency led to a significant

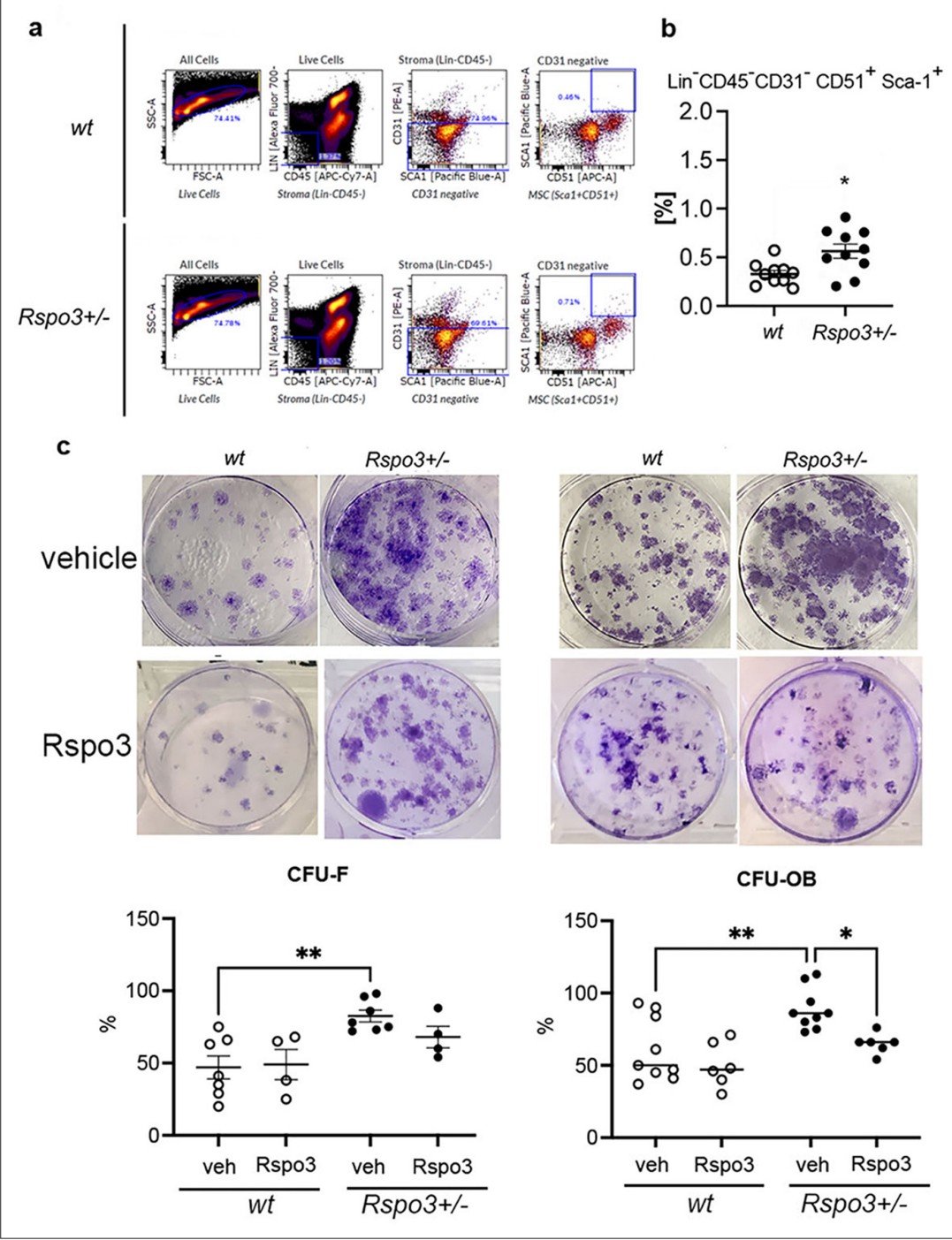

**Figure 2.** *Rspo3* haplo-insufficiency increases the % of osteoprogenitors. (**a**) Representative images of Flow citometry analysis. (**b**) Quantification of the % of Lin-Cd45-Cd31-CD51+Sca+ cells in WT and *Rspo3+/-* bone marrow. Data show all samples and the ± SEM (n=10) *=p< 0.05 by unpaired Student's t-test. (**c**) Representative images of CFU-F and CFU-OB assay and quantification in WT and *Rspo3+/-* mice treated in the absence and presence of Rspo3. Data show all samples and the mean ± SEM (n=4–9) *=p<0.05, **=p<0.005 by two-way ANOVA followed by Fisher's LSD test.

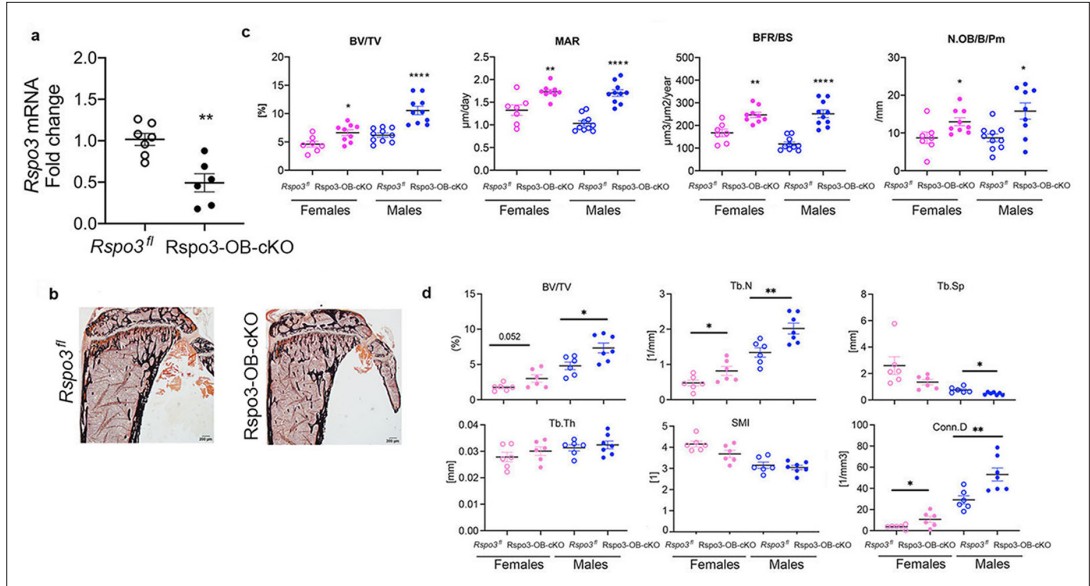

**Figure 3.** Appendicular skeletal phenotype of mice with *Rspo3* targeted deletion in Runx2⁺ cells (Rspo3-OB-cKO). (**a**) *Rspo3* expression in marrow depleted long bones, showing deletion efficiency (n=6–7). Data show all samples and are the mean ± SEM. \*\*p=0.0019 by unpaired Student's t-test. (**b**) Representative images of Von Kossa staining in 8 wk old *Rspo3ᶠˡ* and Rspo3-OB-cKO tibiae. Scale bars = 200 μm (**c**) BV/TV, MAR, BFR/BS, and N.Ob/BPm by histomorphometric analysis in *Rspo3ᶠˡ* and Rspo3-OB-cKO females and males (n=7–10). (**d**) Structural parameters by μCT analysis *Rspo3ᶠˡ* and Rspo3-OB-cKO femur (n=6–7). Data show all samples and the mean ± SEM \*=p<0.05, \*\*=p<0.005, \*\*\*\*=p<0.0001 compared with the correspondent *Rspo3ᶠˡ* group by unpaired Student's t-test. Open circles = *Rspo3* fl and filled circles = Rspo3-OB-cKO.

The online version of this article includes the following figure supplement(s) for figure 3:

**Figure supplement 1.** Axial (L5) and appendicular (femur) skeletal phenotype of *Runx2-Cre,* WT and *Rspo3ᶠˡ mice.*

---

increase in the expression of *Axin2* and *Dkk1,* but not *Sost* in marrow depleted long bone cortical bone (*Figure 5c*). Thus, surprisingly but consistent with the bone and OB phenotypes, *Rspo3* haplo-insufficiency leads to activation of canonical Wnt signaling.

To exclude any function of the remaining Rspo3 on canonical Wnt signaling in the *Rspo3⁺/⁻* mice and cells, we generated *Rspo3⁻/⁻* mouse embryonic fibroblasts (MEFs) at E9.5, before embryonic lethality (*Aoki et al., 2007*; *Kazanskaya et al., 2008*). As shown by several groups (*de Lau et al., 2014*; *de Lau et al., 2012*; *Raslan and Yoon, 2019*; *Park et al., 2018*), we confirmed that while Rspo3 does not activate Wnt signaling by itself, it potentiates exogenous Wnt3a action in WT MEFs as indicated by the Tcf7/Lef luciferase reporter assay (*Figure 6—figure supplement 1a*). However, similar to what is observed in *Rspo3* haplo-insufficient BMSCs and bone marrow depleted long bones cortical bone, at steady-state *Rspo3* deficiency (p<0.0001) (*Figure 6—figure supplement 1b*) led to a marked increase in the expression of the canonical Wnt target genes *Tcf7* (p<0.0001) *and Axin2* (p=0.0017) as well as to a significant increase in the levels of phorphorylated Lrp6 (pLrp6), activated β-catenin and Tcf1 (*Figure 6a and b*). Accordingly, at steady state TOPflash reporter activity was significantly higher in *Rspo3* null MEFs compared with WT MEFs (*Figure 6a*). Upon Wnt3a treatment, as expected, *Axin2,* and *Tcf7* gene expression as well as pLrp6, activated β-catenin levels and TOPflash reporter activity were increased in WT MEFs but were significantly higher in *Rspo3* null MEFs (*Figure 6a and b*). Thus, haplo-insufficiency and absence of *Rspo3* in BMSCs and MEFs led to β-catenin stabilization and enhancement of β-catenin-dependent signaling. We then explored the mechanisms by which this might occur.

## *Rspo3* haplo-insufficiency and deletion impair Dkk1-Wnt inhibitory activity

Activation of canonical Wnt signaling results from changes in endogenous activators and/or inhibitors levels and/or their activity. Interestingly, we found that Wnt3a significantly decreased the expression

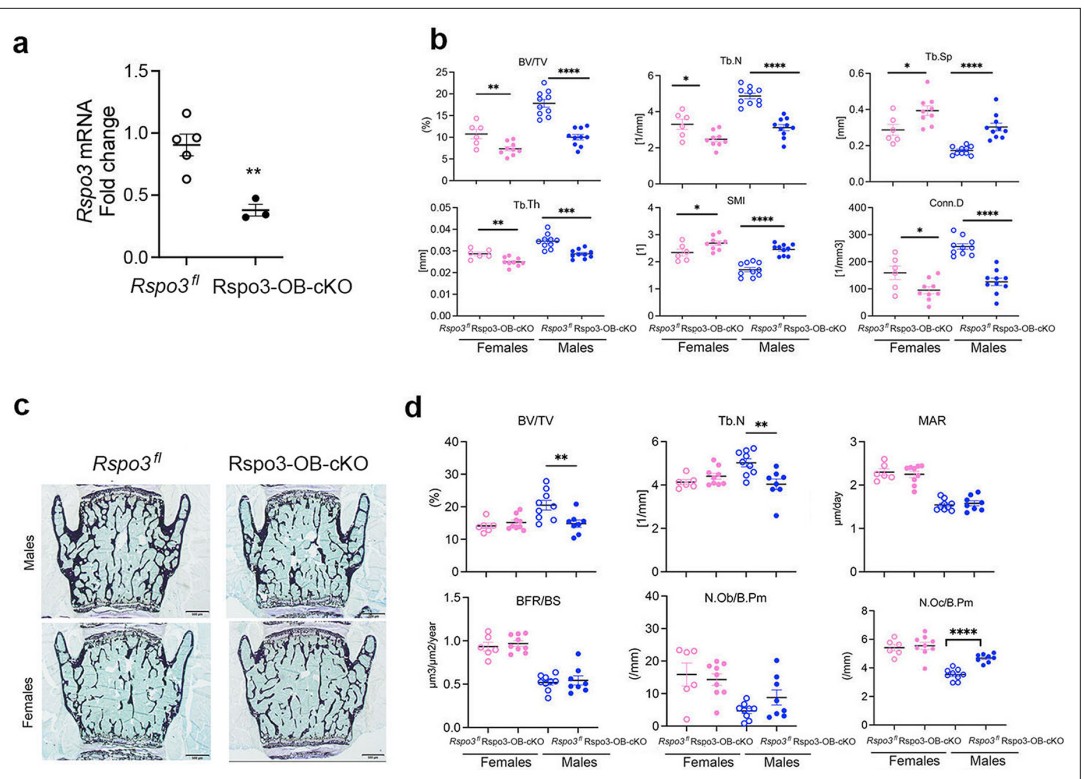

**Figure 4.** Axial skeletal phenotype of mice with *Rspo3* targeted deletion in Runx2+ cells (Rspo3-OB-cKO). (**a**) *Rspo3* expression in vertebrae, showing deletion efficiency (n=3–5). Data show all samples and are the mean ± SEM **p=0.005 by unpaired Student's t-test. (**b**) Structural parameters by µCT analysis *Rspo3^fl^* and Rspo3-OB-cKO L5 vertebrae (n=7–10). (**c**) Representative images of Von Kossa staining in 8 wk old *Rspo3^fl^* and Rspo3-OB-cKO L5 vertebrae. (n=7-10). Scale bars = 500 µm (**d**) BV/TV, Tb.N, MAR, BFR/BS, N.OC/BPm, and N.Ob/BPm by histomorphometric analysis in *Rspo3^fl^* and Rspo3-OB-cKO females and males (n=7–10). Data show all samples and the mean ± SEM *=p<0.05, **=p<0.005, ****=p<0.0001 compared with the correspondent *Rspo3^fl^* group by unpaired Student's t-test. Open circles = *Rspo3* fl and filled circles = Rspo3-OB-cKO.

of *Rspo3* in WT MEFs, whereas it is significantly increased by Dkk1 (*Figure 7a*). Based on our results with *Rspo3^+/-^* and *Rspo3^-/-^* cells, this raised the possibility that Rspo3 participates in a feedback loop that tones down canonical Wnt activity. As shown in +/-, we found that the ability of Dkk1 to block Wnt3a-dependent activation of canonical Wnt signaling was significantly impaired in the absence of *Rspo3*: whereas in WT MEFs 50% reduction in the reporter activity was achieved by 50 ng/mL Dkk1, a dose of 400 ng/mL Dkk1 (8 x higher) was needed to obtain the same level of inhibition in *Rspo3* null MEFs (*Figure 7b*). Impairment of Dkk1 efficacy in *Rspo3* null MEFs was also confirmed by pLrp6 and β-catenin protein levels (*Figure 7c*). To determine whether this relationship between Rspo3 levels and Dkk1 efficacy was also happening in vivo, we crossed *Rspo3^+/-^* mice with mice expressing high levels of Dkk1 in OBs (*Dkk1-Tg* mice), which exhibit impaired canonical Wnt signaling and low trabecular bone mass due to decreased bone formation (*Li et al., 2006*; *Guo et al., 2010*). Supporting our in vitro data, *Rspo3* haplo-insufficiency counteracted the negative effect of OB-targeted overexpression of Dkk1 on trabecular bone mass (*Figure 7d and e* and *Supplementary file 7*).

### *Rspo3* deletion enhances Erk signaling to increase pLrp6 and stabilize β-catenin

Since both *Rspo3* haplo-insufficiency and its deletion led toβ-catenin stabilization, we asked whether Rspo3 might regulate other signaling pathways which in turn can stabilize β-catenin, stimulating osteoblastogenesis and counteracting Dkk1 efficacy. It has been proposed that Rspo3 binding to Lgr4 inhibits Erk phosphorylation (pErk) (*Lee et al., 2020*; *Zhang et al., 2017*; *Zhou et al., 2007*). We, therefore, investigated whether Erk signaling, known to activate Wnt signaling and to regulate OB

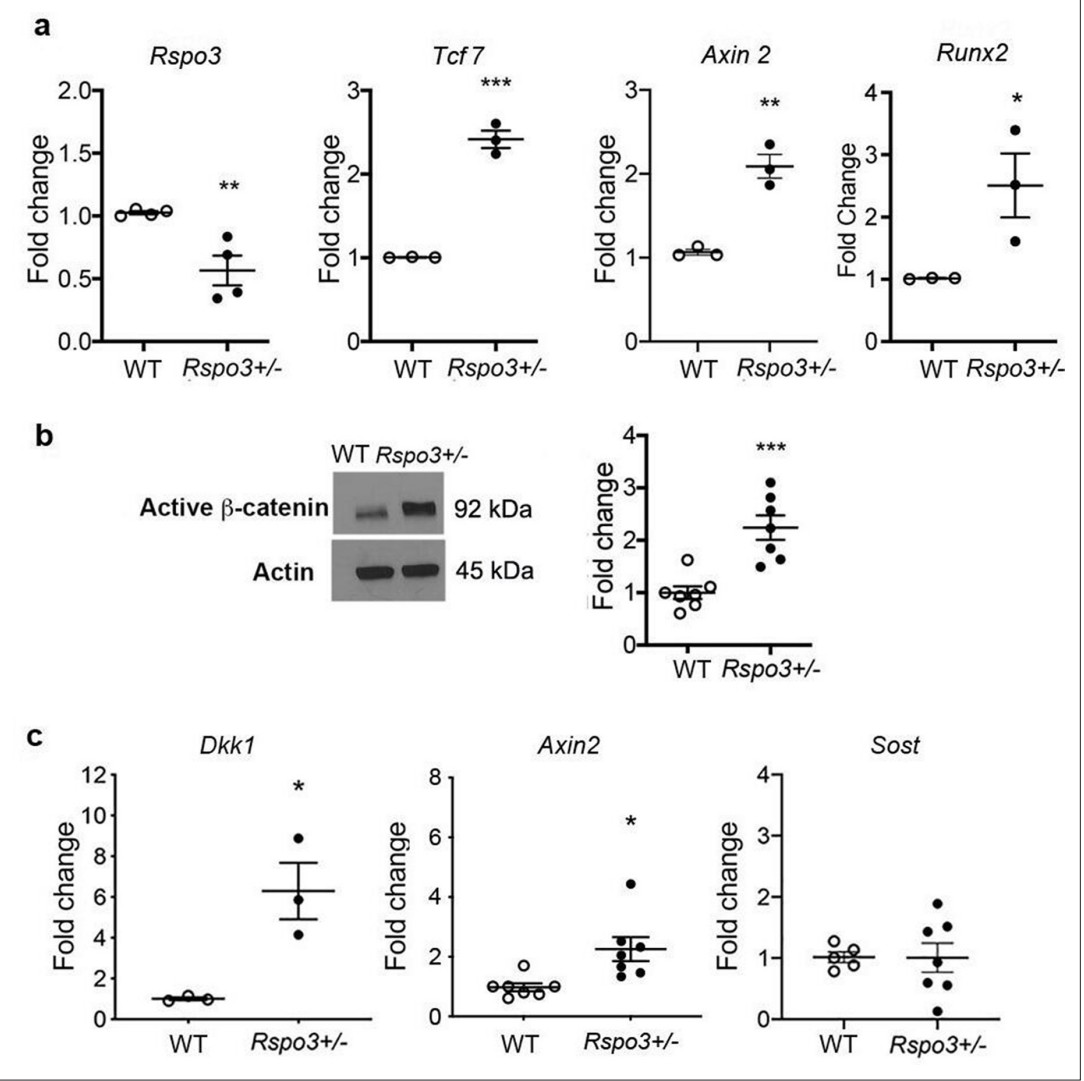

**Figure 5.** *Rspo3* haplo-insufficiency leads to Wnt signaling activation. (**a**) Expression of *Rspo3, Runx2* and selected Wnt target genes in BMSCs (n=3–4). Data show all samples and the mean ± SEM. (**b**) Representative images and quantification of active β-catenin by Western analysis in BMSC isolated from WT and *Rspo3+/-* mice (n=7). (**c**) Expression of selected Wnt target genes in marrow depleted long bones (n=3–7). Data show all samples and the mean ± SEM. *p<0.05, **p<0.005, ***p<0.0005 by unpaired Student's t-test.

The online version of this article includes the following source data for figure 5:

**Source data 1.** *Rspo3* haplo-insufficiency leads to Wnt signaling activation.

**Source data 2.** *Rspo3* haplo-insufficiency leads to Wnt signaling activation.

differentiation and bone mass (*Červenka et al., 2011*; *Gortazar et al., 2013*; *Kim et al., 2019*; *Krejci et al., 2012*), was affected by the absence of Rspo3 in MEFs. *Rspo3* deficiency led to a clear and significant increase in pErk basal levels (*Figure 8a*). Inhibition of pErk by the specific Erk inhibitor U0126, in *Rspo3-/-* cells led to a significant decrease in active β-catenin levels in both steady state and Wnt3a-stimulated cultures. A similar effect was also seen for the levels of pLrp6 (*Figure 8a*). Confirming these findings, the increase in the expression of the canonical Wnt signaling target genes *Tcf7* and *Axin2* in Rspo3 null MEFs was also partially rescued by blocking Erk signaling (*Figure 8b*). In contrast, the Erk inhibitor U0126 significantly decreased both the basal and the Wnt3a-dependent increase in pErk in WT cells but failed to affect significantly active β-catenin levels. Thus, the stabilization of β-catenin seen in the absence of *Rspo3* is due, at least in part, to activation of the Erk pathway.

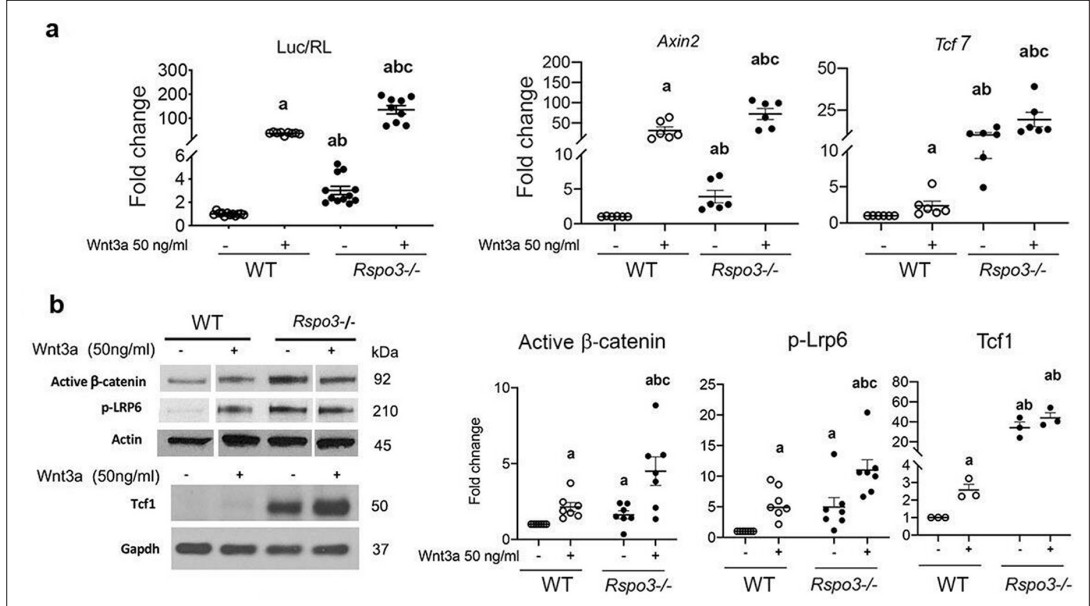

**Figure 6.** *Rspo3* deletion leads to Wnt signaling activation in vitro. (**a**) Luciferase assay and Wnt target gene expression in WT and *Rspo*-/- MEFs treated w/wo Wnt3a (n=6–11). Data show all samples and the mean ± SEM. (**b**) Representative images and quantification of active β–catenin, pLrp6 and Tcf1 by Western analysis in WT and *Rspo*-/- MEFs treated w/wo Wnt3a (n=3–7). Data show all samples and the mean ± SEM. a=p<0.05 vs vehicle WT, b=p<0.05 vs Wnt3a–treated WT and c=p<0.05 vs Wnt3a treated *Rspo3*-/- by two-way ANOVA followed by Fisher's LSD test.

The online version of this article includes the following source data and figure supplement(s) for figure 6:

**Source data 1.** *Rspo3* deletion leads to Wnt signaling activation in vitro.

**Source data 2.** *Rspo3* deletion leads to Wnt signaling activation in vitro.

**Source data 3.** *Rspo3* deletion leads to Wnt signaling activation in vitro.

**Source data 4.** *Rspo3* deletion leads to Wnt signaling activation in vitro.

**Source data 5.** *Rspo3* deletion leads to Wnt signaling activation in vitro.

**Figure supplement 1.** Wnt3a and Rspo3 treatment in WT MEFs.

Consistent with these findings, the increase in CFU-F and CFU-OB formation seen in *Rspo3*+/- BMSCs was significantly decreased by blocking Erk signaling (*Figure 9*). Not surprisingly, knowing the role of Erk signaling activation in osteoblastogenesis, a significant decrease in CFU-OB, but not in CFU-F, formation was also seen in WT cells treated with U0126 (*Figure 9*).

## Discussion

Wnt signaling is central to skeletal development and homeostasis in health and disease (*Baron and Kneissel, 2013*). Understanding the biological mechanisms by which this signal operates is, therefore, of both scientific and clinical interest. R-Spondins, classically considered as positive modulators of Wnt signaling, play an important role in normal development of several tissues and organs, including bone, and are implicated in human diseases (*de Lau et al., 2012*; *Knight and Hankenson, 2014*; *Nagano, 2019*; *Raslan and Yoon, 2019*; *Shi et al., 2016*). Among the 4 R-Spondins, GWAS studies in humans have shown that RSPO3 might be specifically involved in bone homeostasis due to the strong association between RSPO3 common variants and bone mineral density and fracture rate (*Duncan et al., 2011*; *Estrada et al., 2012*; *Medina-Gomez et al., 2012*; *Richards et al., 2008*; *Richards et al., 2012*; *Nilsson et al., 2022*). Our results demonstrate, through several independent lines of genetic in vivo and in vitro experiments, that, counter-intuitively, decreasing *Rspo3* levels globally or specifically in *Runx2*+ OB precursors leads to increased trabecular bone formation and high bone mass in the appendicular skeleton, mainly driven by increased number of OB progenitors and OBs as well as an increase

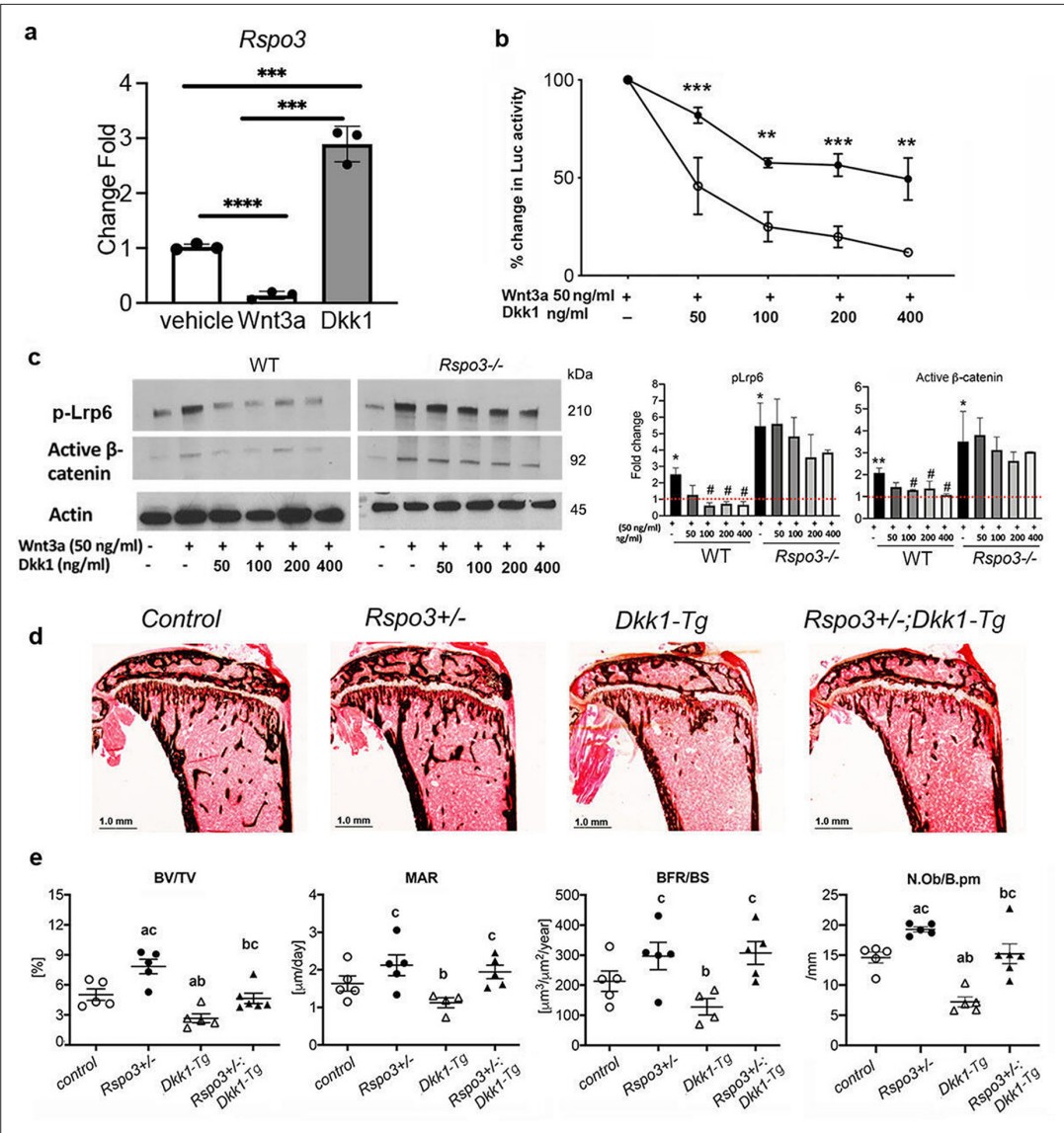

**Figure 7.** *Rspo3* deletion/ haplo-insufficiency impairs Dkk1 efficacy. (**a**) Regulation of Rspo3 by Wnt3a and Dkk1 in WT MEFs (n=3) Data are the mean ± SEM. \*\*\*p<0.0005, \*\*\*\*p<0.0001, by Student's t-test. (**b**) Luciferase assay in WT and *Rspo*[-/-] MEFs treated w/wo Wnt3a and increasing doses of Dkk1 (n=3–4). Data are the mean ± SEM \*\*p<0.005, \*\*\*<p< 0.0005 compared with vehicle same genotype by unpaired Student's t-test. (**c**) Representative images and quantification of active β-catenin and pLrp6 by Western analysis in WT and *Rspo3*[-/-] MEFs treated w/ wo Wnt3a and increasing doses of Dkk1 (n=3). Data are the mean ± SEM \*p<0.05, \*\*p<0.005 vs WT vehicle, # p<0.05, vs Wnt3a–treated same genotype by unpaired Student's t-test. (**d**) Representative images of Von Kossa staining in 6 wk old female mice (n=5-6). Scale bars = 1.0 mm. (**e**) BV/TV, MAR, BFR/BS, and N.Ob./B.pm by histomorphometric analysis in females (n=5–6). Data show all samples and the mean ± SEM a=p<0.05 compared with *control* mice, b=p < 0.05 compared with *Rspo3*[+/-] mice, c=p<0.05 compared with *Dkk1-Tg* mice by two-way ANOVA followed by Fisher's LSD test.

The online version of this article includes the following source data for figure 7:

**Source data 1.** *Rspo3* deletion/ haplo-insufficiency impairs Dkk1 efficacy.

**Source data 2.** *Rspo3* deletion/ haplo-insufficiency impairs Dkk1 efficacy.

**Source data 3.** *Rspo3* deletion/ haplo-insufficiency impairs Dkk1 efficacy.

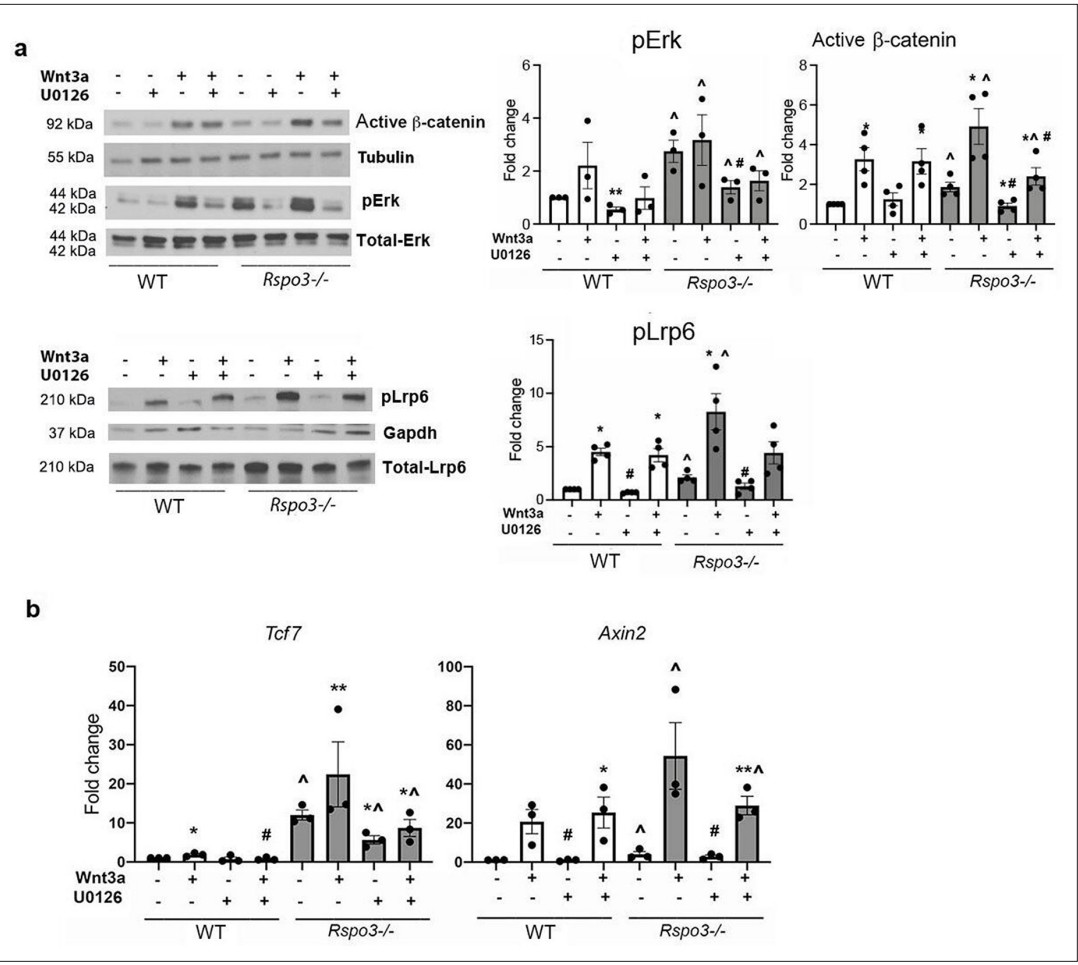

**Figure 8.** Erk signaling is involved in the Wnt signaling activation seen in the absence of Rspo3. (**a**) Representative images and quantification of pERK, active β-catenin and pLrp6 levels by western analysis in WT and *Rspo3⁻/⁻* MEFs treated w/wo w/wo Wnt3a and U0126. (**b**) Expression of Wnt target genes in WT and *Rspo3⁻/⁻* MEFs treated w/wo Wnt3a and U0126. Data show all samples and the mean ± SEM (n=3–4) *p<0.05, **p<0.005 vs vehicle of the same genotype, ^=p<0.05 vs WT vehicle and # p<0.05 vs Wnt3a–treated same genotype by unpaired Student's t-test.

The online version of this article includes the following source data for figure 8:

**Source data 1.** Erk signaling is involved in the Wnt signaling activation seen in the absence of Rspo3.

**Source data 2.** Erk signaling is involved in the Wnt signaling activation seen in the absence of Rspo3.

**Source data 3.** Erk signaling is involved in the Wnt signaling activation seen in the absence of Rspo3.

**Source data 4.** Erk signaling is involved in the Wnt signaling activation seen in the absence of Rspo3.

**Source data 5.** Erk signaling is involved in the Wnt signaling activation seen in the absence of Rspo3.

**Source data 6.** Erk signaling is involved in the Wnt signaling activation seen in the absence of Rspo3.

**Source data 7.** Erk signaling is involved in the Wnt signaling activation seen in the absence of Rspo3.

in their bone forming activity. Unexpectedly though, and as recently reported by *Nilsson et al., 2021*, we also found that while the axial skeleton is only moderately affected in the haplo-insufficient mice, vertebrae are affected in the opposite manner, that is with a decrease in trabecular bone mass, after OB precursor-targeted deletion of *Rspo3*. Although it is not possible to identify the mechanism(s) responsible for these differential responses of the skeleton, it is worth mentioning here that axial and appendicular skeleton have different embryonic origin (paraxial mesoderm and later plate mesoderm, respectively) (*Berendsen and Olsen, 2015*) and that a new population of skeletal stem cells has been recently identified in the vertebrae (vSSC) that is absent from the long bones (*Ay et al., 2022*), raising the possibility that distinct populations of stem cells and non-stem progenitors in the vertebrae and in

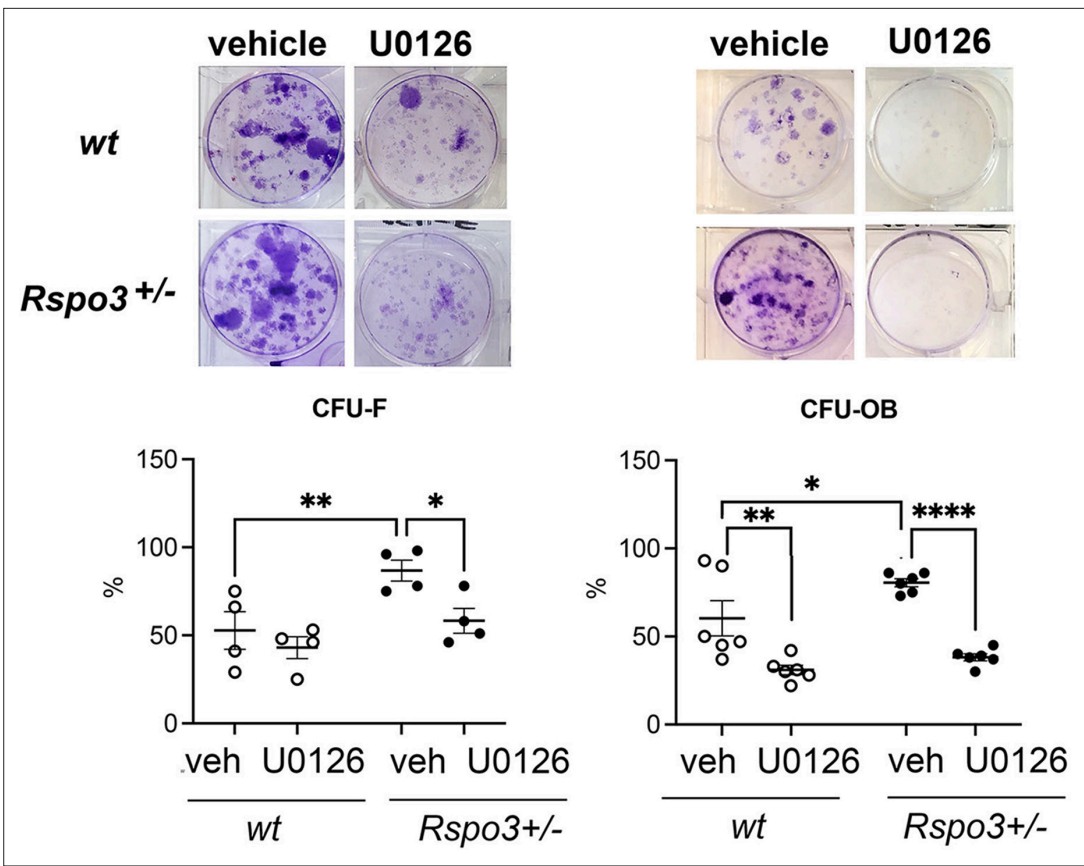

**Figure 9.** Effect of Erk signaling inhibition of CFU assays. Representative images of CFU-F and CFU-OB and quantification in WT and *Rspo3*[+/-] mice treated with and w/o U0126. Data show all samples and the mean ± SEM (n=3) *=p<0.05, **=p<0.005, ****=p<0.0001 two-way ANOVA followed by Fisher's LSD test.

the long bones might respond differentially to stimuli. These findings underline the complexity of Wnt signaling regulation of bone homeostasis and support the hypothesis that different bone microenvironments, their embryonic origin, their differential exposure and response to mechanical cues, and the differential expression levels of receptors and co-receptors as well as of agonists and antagonists that balance Wnt signaling activity, might induce specific and distinct skeletal responses. Nevertheless, our findings show that skeletal phenotypes should always be investigated in both the appendicular and axial skeleton, as these two regions may respond differently. In contrast, Nilsson et al. performed their studies only in vertebrae. Furthermore, this is a trabecular bone-rich skeletal site, but the DEXA studies that led to the identification of a link between RSPO3 variants and bone fragility in humans were performed in the distal forearm, a cortical-rich appendicular skeletal site (*Nilsson et al., 2021*; *Nilsson et al., 2022*). Furthermore, both groups found that cortical bone is not significantly affected by *Rspo3* haplo-insufficiency or deletion in mice, confirming that trabecular and cortical bone are differentially regulated (*Movérare-Skrtic et al., 2014*; *Kiper et al., 2016*) and further underlining the biologic complexity of Wnt signaling regulation of bone homeostasis.

Thus, our studies reveal a novel and unexpected role of Rspo3 in the Wnt signaling machinery and bone homeostasis. Given the unexpected nature of our observations it is important to summarize here the multiple independent in vivo and in vitro experiments that unequivocally support the novel concept that decreasing the levels of Rspo3 can, at least in vitro and in the cellular environment of the appendicular skeleton's trabecular bone, result in the activation of canonical Wnt signaling, increasing osteoblast differentiation and bone formation, and thereby bone mass: 1/Global haplo-insufficient or *Runx2-Cre* driven depletion of *Rspo3* increased bone mass, OB numbers, and BFR in mice long bones; 2/Ex vivo, haplo-insufficient BMSCs exhibited increased OB differentiation and canonical Wnt signaling; 3/*Rspo3* null MEFs also exhibited an enhanced Wnt signaling activity; 4/Basal levels of Erk phosphorylation were high in these cells and inhibition of Erk phosphorylation prevented

the activation of Wnt signaling; 5/*Rspo3* haplo-insufficiency rescued partially the inhibition of Wnt signaling induced by Dkk1 in vitro and in vivo.

In vitro studies have demonstrated that overexpression of and treatment with Rspo1 or Rspo2 enhance Wnt ligand-mediated OB differentiation (*Knight and Hankenson, 2014*; *Lu et al., 2008*; *Knight et al., 2018*). The literature and our own in vitro studies confirm that Rspo3 can be, as expected, a co-activator of canonical Wnt signaling in the Topflash assay, potentiating Wnt3a-dependent activation of canonical Wnt signaling in cellular assays (*Yoon and Lee, 2012* and *Figure 6—figure supplement 1*). However, the expression of canonical Wnt target genes and the levels of pLrp6, activated β-catenin and Tcf1 were also markedly increased in *Rspo3* haplo-insufficient BMSCs and in *Rspo3*-null MEFs, independent of the addition of Wnt3a to the assays, indicating that, counter-intuitively, the decrease or absence of *Rspo3* activates mechanisms that favor β-catenin-dependent signaling. This is in contrast with the data reported by Nilsson et al. who show a decrease in Dkk1 and Sost, but this was done in the vertebrae of *Runx2-CreRspo3^{fl/fl}* mice, where BFR is in fact decreased, or a decrease in *Tcf7* and *Lef1* expression in calvarial OBs (cortical-derived cells *Azzolin et al., 2014*) treated with and without Tamoxifen to induce *Rspo3* deletion in vitro (*Nilsson et al., 2021*), when cortical bone is in fact not affected. Although we have not analyzed mRNA expression in the vertebrae or calvaria, our extensive set of data (Topflash, Q-RTPCR, Western blot analysis for several Wnt signaling downstream proteins) using different cells (BMSC and MEFs) and long bones clearly show an activation of Wnt signaling with *Rspo3* haploinsufficiency or deletion.

Another new finding from our studies is that *Rspo3* expression is strongly repressed by Wnt3a and increased by Dkk1. This, together with the results discussed above, suggests that Rspo3 may provide a negative feedback-loop helping to balance canonical Wnt activity. Dkk1 efficacy in blocking Wnt3a-dependent activation of canonical Wnt signaling is significantly impaired in the absence of *Rspo3* and *Rspo3* haplo-insufficiency antagonizes the inhibition of bone formation induced by OB-targeted expression of Dkk1, confirming in vivo that *Rspo3* haplo-insufficiency counteracts the OB-Dkk1 dependent function. Our results show that this is the result of intracellular changes in alternative pathways regulated by Rspo3. Indeed, we found that the Erk signaling pathway is activated and the

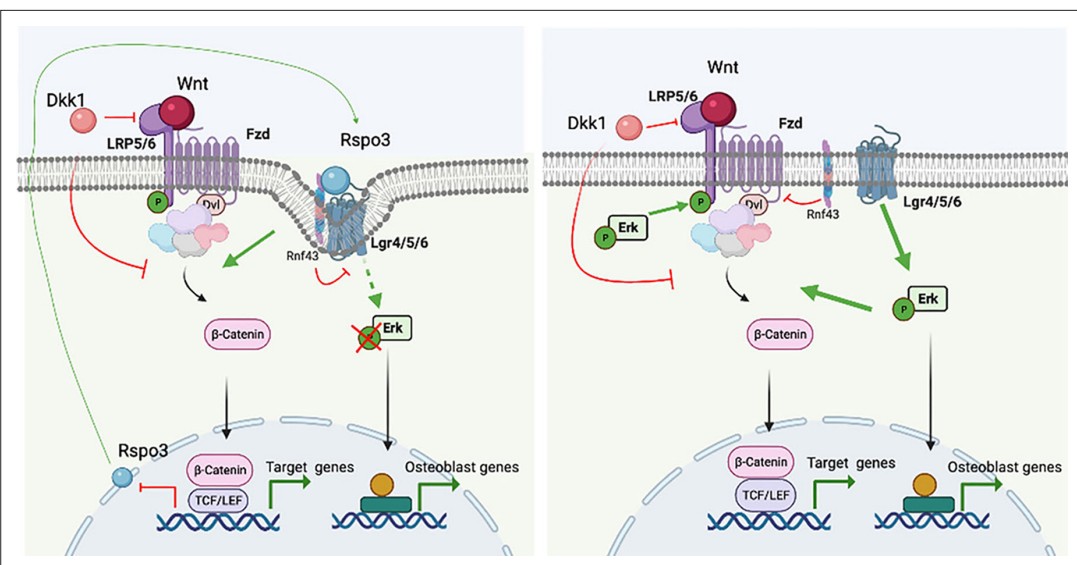

**Figure 10.** Proposed model. Rspo3 has a dual mode of action to regulate canonical Wnt signaling and thereby bone formation. This duality is based on the regulation of two distinct signaling cascades and their crosstalk: Rspo3 functions via both the Lgr/Rnf43/Znrf3 and the Lgr/Erk axes. In the presence of Rspo3, the Rspo3/Lgr/Rnf43/Znrf3 axis boosts Wnt signaling strengths by the membrane clearance of Rnf43/Znrf3 and subsequent stabilization of Fzd receptors. In addition, binding of Rspo3 to Lgr impairs Erk signaling likely due to the membrane clearance of the Lgr/Rnf43/Znrf3 receptors, preventing Erk signaling activation. Deletion of Rspo3 would dampen Wnt signaling at the cell surface by preventing the Rnf43/Znrf3 effects while promoting Erk activation downstream of Lgr receptors in turn enhancing Lrp5/6 phosphorylation and β–catenin stabilization intracellularly, which has a more potent effect and overcompensates the decrease in Rspo3-dependent proximal Wnt activation in osteoblasts and their progenitors. Figure created with Biorender.

basal level of Lrp6 phosphorylation enhanced by *Rspo3* depletion. Thus, the activation of the Wnt signaling pathway when Rspo3 is expressed at low levels results from intra-cellular changes, distal to the receptor complexes. Our data and that of others (*Lee et al., 2020*) suggest that there are alternate Rspo3-mediated signaling mechanisms, separate from the Fzd/Lrp/β-catenin Wnt pathway, including the Wnt/PCP signaling (*Glinka et al., 2011*; *Hao et al., 2012*) and that these events can in turn regulate Wnt signaling intra-cellularly, such that β-catenin can be stabilized independent of the proximal activation of the canonical Wnt signaling machinery through changes in other signaling pathways (*Baron and Kneissel, 2013*; *Krejci et al., 2012*; *Azzolin et al., 2014*).

Thus, deletion of *Rspo3* enhances Erk signaling which, in turn, stabilizes β–catenin independent of the canonical Wnt signaling receptor complex. In turn, this has a positive effect on OB differenti-ation (*Červenka et al., 2011*; *Kim et al., 2019*; *Krejci et al., 2012*). Our finding that the increase in Wnt signaling activation and the OB potential of MSCs seen in the absence of *Rspo3* is abrogated by blocking Erk signaling confirms that the activation of Erk signaling associated with *Rspo3* deficiency is responsible, at least in part, for the observed Wnt signaling activation.

Supporting our findings, in vitro studies have shown that *Rspo3* silencing activates Erk signaling downstream of Lgr4, leading to increased OB differentiation of human adipose-derived stem cells (*Zhang et al., 2017*). This study did not however establish a connection between Erk and Wnt signaling. Although Lgrs function as receptors for Rspos, and Rspos/Lgrs interactions enhance Wnt signaling by inducing the clearance of Rnf43 and Znrf3 (*Ruffner et al., 2012*; *Wang et al., 2013*; *de Lau et al., 2011*), there is also strong evidence that Rspos/Lgrs interaction can activate distinct signaling cascades that can affect bone, including the cAMP/PKA/Creb signaling pathway in Lgr4 null mice (*Luo et al., 2009*) and the Erk signaling cascade (*Zhang et al., 2017*; *Xu et al., 2016*; *Lin et al., 2019*; *Vieira et al., 2015*).

Based on our observations, we propose that Rspo3 has a dual mode of action to regulate canonical Wnt signaling and bone formation. This duality is based on the regulation of two distinct signaling cascades and their crosstalk: Rspo3 functions via both the Lgr/Rnf43/Znrf3 and the Lgr/Erk axes, and while activation of the Lgr/Rnf43/Znrf3 axis boosts Wnt signaling strength by the membrane clear-ance of Rnf43/Znrf3 and subsequent stabilization of Fzd receptors, binding of Rspo3 to Lgr impairs Erk signaling, preventing Erk signaling activation and further stabilization of β–catenin (*Figure 10*). Thus, haplo-insufficiency and deletion of Rspo3 would dampen Wnt signaling at the cell surface by preventing the Rnf43/Znrf3 effects while intracellularly enhancing pLrp6 and β–catenin stabilization, via Erk phosphorylation, overcompensating the decrease in Rspo3-dependent Lrp5/6 receptors-dependent Wnt activation in OBs and their progenitors. Because activation of the Lgr/Rnf43/Znrf3 cascade is not exclusively dependent on Rspo3, deletion of *Rspo3* would only hinder canonical Wnt signaling partially. In contrast, lack of *Rspo3* promotes the Lgr/Erk cascade, to not only enhance β–catenin stabilization (*Figure 10*) but also regulate OB differentiation and bone formation. This model also explains the observed loss of Dkk1 efficacy in inhibiting Wnt signaling: the Erk-dependent stabilization of β–catenin being independent of Wnt receptor activation, Dkk1, which binds to the LRP5/6 receptors, cannot dampen the activation of downstream events as they are independent of the LRP5/6-Fzd receptor complex.

Supporting the fact that Rspo3 can also regulate Wnt-independent pathways, a recent study has suggested that Rspo3 acts as an antagonist to BMPR1A, inhibiting BMP signaling during development (*Lee et al., 2020*). Thus, our observations may be due, at least in part, to changes (activation) in BMP signaling, which in turn could lead to the observed increase in pErk (*Zhou et al., 2007*). Although this remains a possibility, it seems unlikely. First, in contrast to our observations here, BMP activation in the adult skeleton has been linked to activation of non-canonical Wnt signaling, increased Sost expression, and bone resorption (*Kiper et al., 2016*; *Kamiya et al., 2016*). Second, several studies have shown that activation of BMP signaling in the osteoblast lineage has a negative impact on bone formation and bone mass (*Kamiya et al., 2010*; *Kamiya et al., 2008*; *Ko et al., 2017*; *Shi et al., 2018*).

In conclusion, our studies indicate that Rspo3 is required for skeletal homeostasis and show for the first time that axial and appendicular skeleton are differentially regulated by this Wnt signaling regu-lator. The molecular basis for this differential regulation has not been elucidated but might include different embryologic origin, differences in local micro-environment, or differences in mechanical loading, for instance. Nevertheless, our in vitro studies suggest that Rspo3 regulates bone formation through its interaction not only with the Wnt receptor machinery, as a positive co-activator, but also

with other signaling pathways that affect β–catenin stability independent of the receptor complexes. Consequently, its deletion removes a co-activator of Wnt signaling, potentially decreasing bone formation, but also promotes Erk signaling activation, increasing β–catenin stability sufficiently to enhance bone formation and increase bone mass in the appendicular skeleton. Furthermore, because *Rspo3* depletion increases *Dkk1* and Dkk1 increases *Rspo3* expression, this study also reveals a novel Rspo3-dependent negative feedback Wnt signaling regulatory loop. These findings have important implications for understanding the pleiotropic functions of Rspos and Wnt signaling in skeletal homeostasis and the mechanisms that regulate bone mass.

# Materials and methods

## Biological variables and reproducibility

To conduct the proposed studies, we insured strict application of scientific methods that supports robust and unbiased design, analysis, interpretation, and reporting of results, and sufficient information for all studies undertaken. In vivo analysis was performed in males and females, at different ages, and in the axial and appendicular skeleton. *Rspo3*$^{+/-}$ mice and Rspo3-OB-cKO mice with targeted deletion of *Rspo3* in osteoprogenitors (*Runx2-Cre*) were investigated. Given that our studies with *Rspo3* mutant mice do not identify sex differences, and the osteoblast specific Dkk1 transgenic (*Dkk1-Tg*) mice did not show sex differences (*Guo et al., 2010*), studies including crossing with the two strains were performed in female mice. To avoid bias, data were collected in a blinded fashion, in that the observer was unaware of the experimental groups and more than one individual performed key studies. In vivo studies were performed with n=4–*10* mice per genotype. We based this number on a priori calculations for power to detect differences in the primary outcome skeletal phenotype by histomorphometric analysis. We found that based on three independent methods for measuring the skeletal phenotype 4–10 mice per group provide enough power (80%–90%) to find statistical significance at p<0.05. Ex vivo and in vitro studies involved at least three biological replicates per group/treatment.

## Animals

*Rspo3*$^{+/-}$ *and Rspo3*$^{fl}$ mice were provided by Dr. Christof Niehrs (DKFZ-ZMBH Alliance, Germany) and were previously described (*Kazanskaya et al., 2008*). The osteoblast specific Dkk1 transgenic (*Dkk1-Tg*) mice, expressing high levels of Dkk1 in osteoblasts, were generously provided by Dr. Guo and Dr. Kronenberg (Massachusetts General Hospital, MA, USA) (*Guo et al., 2010*). *Runx2-Cre* mice were provided by Dr. Tuckermann (Ulm University, Ulm, Germany) (*Rauch et al., 2010*). All experiments were performed with age- and sex-matched littermates. All animals are in the C57BL/6 background and were housed in the Harvard Center for Comparative Medicine and all experimental procedures were approved by the Harvard University Institutional Animal Care and Use Committee. The protocol number associated with the ethical approval of the animal work is IS1045.

## Skeletal phenotype

For bone histomorphometric analysis of *Rspo3*$^{+/-}$ mice, 6, 12, or 18 wk-old mice were injected with 20 mg/kg of calcein and 40 mg/kg of demeclocycline (Sigma Aldrich, St. Louis, MO, USA) 6, 8, or 9 days, respectively and 2 days prior to the sacrifice. For mice with *Rspo3* deletion in Runx2$^{+}$ cells, 8-wk-old mice were injected with 20 mg/kg of calcein and 40 mg/kg of demeclocycline, 8 and 2 days prior to sacrifice and for the experiments with the *Dkk1-Tg* mice, 6-wk-old mice were injected 6 and 2 days prior to sacrifice. Bone histomorphometric analysis was performed within the proximal tibia under 200× magnification in a 0.9 mm high and 1.3-mm-wide region that was 200 μm away from the growth plate. For vertebra analysis, data were obtained under 200× magnification in a 1.3 mm × 1.8 mm region away from the growth plate to avoid including primary spongiosa. Consecutive sections of the proximal tibia and frontal sections of the vertebral body (4 μm thickness) were stained with von Kossa and Toluidine blue for the analysis of cellular parameters and osteoid. Bone sections were viewed with a Nikon E800 microscope equipped with Olympus DP71 digital camera (RRID:SCR_020326). Images were captured using Olympus CellSens software (RRID:SCR_014551). The OsteoMeasure analyzing software (Osteometrics) was used to generate and calculate the data. Structural parameters bone volume fraction (BV/TV), trabecular thickness (Tb.Th), trabecular number

(Tb.N), and trabecular separation (Tb.Sp) were obtained by calculating the average of two different measurements from consecutive sections. The structural, dynamic, and cellular parameters were presented according to the standardized nomenclature (*Dempster et al., 2013*).

A high-resolution desktop micro-tomographic imaging system (μCT40, Scanco Medical AG, RRID:SCR_017119, Brüttisellen, Switzerland) was used to assess trabecular bone architecture and mineral density in the L5 vertebral body and femur and cortical bone architecture in the diaphysis of the femur or the tibia. Scans were acquired using a 10 μm$^3$ isotropic voxel size, 70 kVP, 114 mAs, and 200ms integration time. μCT scanning and analysis were performed according to recommended guidelines (*Bouxsein et al., 2010*). Trabecular bone within this region was segmented from soft tissue using a threshold of 400 mg HA/cm$^3$. The Scanco trabecular morphology evaluation script was used to measure trabecular bone volume fraction (Tb.BV/TV, %), trabecular bone mineral density (Tb.BMD, mg HA/cm$^3$), Tb.N (mm$^{-1}$), Tb.Th (μm), Tb.Sp (μm), structural model index (SMI), and connectivity density (Conn.D, mm$^{-3}$). Cortical bone was assessed in 500 μm long regions (50 transverse slices) at the femoral mid-diaphysis and tibial diaphysis (2 mm superior to the distal tibiofibular junction). Images within the cortical region of interest were segmented using a threshold of 700 mgHA/cm$^3$ and then the Scanco midshaft evaluation script was used to measure total cross-sectional area Total area (Tt.Ar, mm$^2$), cortical bone area (Ct.Ar, mm$^2$), medullary area (Ma.Ar, mm$^2$), bone area fraction (Ct.Ar/Tt.Ar, %), cortical tissue mineral density (Ct.TMD, mgHA/cm$^3$), cortical thickness (Ct.Th, mm).

## Flow cytometry

Bone marrow was analyzed by flow cytometry as previously described (*Schepers et al., 2013*). Briefly, bone marrow cells were flushed from femurs and tibiae of 6–8 wk old WT or *Rspo3*$^{+/-}$ mice and washed with Hank's Balanced Salt Solution (HBSS). Residual bone samples were further digested in 3 mg/ml type I collagenase (Worthington Biochemical Corp., Lakewood, NJ 08701, USA) for 1 hr at 37 °C and released cells were mixed with flushed bone marrow cells. Cells were stained with LIVE/DEAD Fixable Aqua Dead Cell Stain Kit (Thermo Fisher Scientific, Waltham, MA, USA), AF700-anti-lineage (RRID:AB_2715571), PE-anti-CD31 (RRID:AB_2572182), PB-anti-Sca-1 (RRID:AB_2143237), anti- Cy7APC-CD45 (RRID:AB_2860726), and biotin-anti-CD51 (RRID:AB_313073) with streptavidin-APC antibodies (BioLegend, San Diego, CA, USA). Cells were analyzed on a BD FACS ARIAII (RRID:SCR_018091) upon exclusion of dead cells.

## Bone marrow stromal cells and calvarial osteoblasts

Bone marrow cells were flushed from femurs and tibiae of 6–8 wk-old WT or *Rspo3*$^{+/-}$ mice and cultured in DMEM supplemented with 10% fetal bovine serum (FBS) and 1% penicillin (100 U/ml) and streptomycin (100 μg/ml) for 3 days (GIBCO, Thermo Fisher Scientific, Waltham, MA, USA). Adherent MSCs were counted, re-plated onto at a 5,000 /cm$^2$ density, and RNA or protein isolated 3 days later. For colony forming unit assays, flushed bone marrow stromal cells were plated (3X10$^6$/6 wells) for CFU- (Fibroblast) F and CFU-OB assays. Cells were treated either with recombinant mouse Rspo3 (100 ng/ml) (all from R&D system, Minneapolis, MN, USA) or U0126 (10 μM) (Selleckchem, Houston TX, USA). CFU-F was detected by staining with 0.2% crystal violet in 2% ethanol for 1 hr after 10 days in culture and CFU-OB was detected by alkaline phosphatase activity with Napthol AS-MX, n,n-dimethylformamide and Fast Blue RR salt (Sigma Aldrich, St. Louis, MO, USA) after 12 days in culture with OB differentiation medium: DMEM supplemented with 10%FBS, 1% penicillin (100 U/ml) and streptomycin (100 μg/ml) (GIBCO, Thermo Fisher Scientific, Waltham, MA, USA), 5 μg/ml ascorbic acid and 10 mM β-glycerolphospahte (Sigma-Aldrich, St. Louis, MO, USA). Calvarial OBs were isolated from 1 to 3-day-old pups via serial enzymatic digestions and cultured as previously reported (*Chen et al., 2019*).

## Mouse embryonic fibroblasts (MEFs) primary culture

To obtain WT and *Rspo3*$^{-/-}$ MEFs, *Rspo3*$^{+/-}$ males and females were crossed, and the morning of vaginal plug detection was defined as embryonic day (E) 0.5. At E9.5, whole embryos were isolated, washed in PBS, minced in 0.05% trypsin (GIBCO, Thermo Fisher Scientific, Waltham, MA, USA) followed by incubation at 37 C for 10 min. After incubation samples were pipetted to obtain single cell suspension and cells cultured in DMEM supplemented with 10% FBS and 1% penicillin/streptomycin. All the experiments were performed in passage 4–6 of WT or *Rspo3*$^{-/-}$ MEFs. Cells were treated either

50 or 200 ng/ml of recombinant human Wnt3a, recombinant human Dkk1 (50–400 ng/ml), recombinant human and/or mouse Rspo3 (100 ng/ml) (all from R&D system, Minneapolis, MN, USA) or U0126 (10 µM) (Selleckchem, Houston TX, USA). For TOPflash luciferase reporter assay, cells were transiently co-transfected with 400 ng TOPflash-luc reporter plasmid (RRID:Addgene_12456) and 10 ng control pCMV-Renilla-luciferase (RRID:Addgene_45968, Promega, Madison WI, USA) using Lipofectamine 2000 (Invitrogen, Thermo Fisher Scientific, Waltham, MA, USA) according to the manufacturers protocol. Cells were subjected to serum starvation in DMEM containing 1% FBS overnight. Cells were subsequently treated with recombinant human Wnt3a in the presence and absence of increasing concentration of Dkk1 for 24 hr followed by luciferase assay using the Dual-Glo Assay system (Promega, Madison WI, USA) according to the manufacturers protocol. Data were normalized by Renilla-firefly activity and presented as fold change compared with control group.

## Osteoclast primary culture and mix-matched co-cultures

Murine bone marrow macrophages were isolated from bone marrow flushed tibiae and femurs of WT and *Rspo3*+/- mice at 6–8 wk-old as described previously (*Chen et al., 2019*). Briefly, cells were cultured in complete α-MEM with 30 ng/ml macrophage colony-stimulating factor M-CSF (R&D system, Minneapolis, MN, USA) in suspension culture dish to which stromal cells and lymphoid cells cannot adhere, at 37 °C for 2–3 days. For osteoclast generation, cells were cultured in 30 ng/ml M-CSF and 10 ng/ml RANKL (R&D systems, Minneapolis, MN, USA). For co-culture experiments, mouse calvarial osteoblasts were isolated from newborn WT and *Rspo3*+/- as previously reported (*Movérare-Skrtic et al., 2014*; *Chen et al., 2019*) and seeded in 96-well plates (2.000 cells/well) in complete osteogenic α-MEM containing 100 nM Vitamin D3 and 1 µM prostaglandin E2 (Enzo Life Science, Farmingdale, NY, USA). After 3 days, 10,000 BMM from WT and *Rspo3*+/- mice at 6–8 week-old mice were added per well and cocultured for 9 days in complete osteogenic α-MEM. Tartrate-resistant acid phosphatase (TRAP) staining was performed to evaluate the number of osteoclasts according to the manufacture's protocol (Sigma-Aldrich, St. Louis, MO, USA).

## Western Blot analysis

Five µg of total proteins were resolved by SDS-PAGE under reducing conditions. Immunodetection was performed with antibodies specific to: Active β-catenin, phosphorylated (p) Lrp6, p-Erk, total Erk, Tcf1, Lrp6, and Tubulin [(CST8814, RRID:AB_11127203) (CST2568, RRID:AB_2139327) (CST9101, RRID:AB_331646), (CST9102, RRID:AB_330744), (CST2203, RRID:AB_2199302), (CST3395, RRID:AB_1950408), (CST2125, RRID:AB_2619646) Cell Signaling, Beverly, MA, USA] GAPDH and Actin [(SC32233, RRID:AB_627678) and (SC47778, RRID:AB_626632) Santa Cruz, Santa Cruz, CA, USA]. Immunoreactivities were assessed using ECL plus kit following the manufacture's protocol (Perkin Elmer, Waltham, MA, USA). Quantification was performed using Image J (RRID:SCR_003070) Protein levels were normalized to the levels of housekeeping protein or total protein in within the same sample.

## Quantitative-real time PCR

Total RNA was isolated from cells using the RNeasy Mini Kit (Qiagen Germantown, MD, USA) according to the manufacturer's protocols. Total RNA from cortical bone of WT and *Rspo3*+/- mice was extracted using Trizol reagent (Invitrogen) followed by RNeasy Micro Kit (Qiagen Germantown, MD, USA) according to the manufacturer's protocols. cDNA was synthesized using iScript cDNA synthesis kit (BIO-RAD. Hercules, CA, USA) and quantitative real time PCR performed. mRNA levels encoding each gene of interest were normalized for β2M or actin mRNA in the same sample and the relative expression of the genes of interest was determined using the formula of *Livak and Schmittgen, 2001*. Data are presented as fold change relative to WT cells or animals.

## Statistical analysis

Data are expressed as the mean ± SEM. All experiments include at least three biological replicates and were done in duplicate or triplicate. Values represent the number of biological replicates. Statistical analysis was conducted using unpaired two-tail Student's t-test, or two-way ANOVA followed by post-hoc test for multiple comparisons. GraphPad PRISM 9 (RRID:SCR_002798) was also used

for statistical analysis. A two-sided p-value of <0.05 was considered as the threshold for statistical significance.

## Acknowledgements

This work was supported by NIH-NIAMS R01AR064724 to RB and in part by NIH-NIDCR R01DE029615 to FG.

## Additional information

### Funding

| Funder | Grant reference number | Author |
|---|---|---|
| National Institute of Arthritis and Musculoskeletal and Skin Diseases | R01AR064724 | Roland Baron |
| National Institute of Dental and Craniofacial Research | R01DE029615 | Francesca Gori |

The funders had no role in study design, data collection and interpretation, or the decision to submit the work for publication.

### Author contributions

Kenichi Nagano, Data curation, Formal analysis, Investigation, Methodology, Writing – original draft, Writing – review and editing; Kei Yamana, Data curation, Formal analysis, Methodology, Writing – original draft; Hiroaki Saito, Riku Kiviranta, Data curation, Formal analysis, Investigation, Methodology; Ana Clara Pedroni, Dhairya Raval, Data curation, Formal analysis, Methodology; Christof Niehrs, Resources, Writing – review and editing; Francesca Gori, Conceptualization, Resources, Data curation, Formal analysis, Supervision, Funding acquisition, Investigation, Methodology, Writing – original draft, Project administration, Writing – review and editing; Roland Baron, Conceptualization, Resources, Supervision, Funding acquisition, Validation, Investigation, Writing – original draft, Project administration, Writing – review and editing

### Author ORCIDs

Kenichi Nagano http://orcid.org/0000-0003-3145-4841
Francesca Gori http://orcid.org/0000-0001-5685-8303

### Ethics

This study was performed in strict accordance with the recommendations in the Guide for the Care and Use of Laboratory Animals of the National Institutes of Health. All experiments were performed with age- and sex-matched littermates. All animals are in the C57BL/6 background and were housed in the Harvard Center for Comparative Medicine and all experimental procedures were approved by the Harvard University Institutional Animal Care and Use Committee. The protocol number associated with the ethical approval of the animal work is IS1045.

### Decision letter and Author response

Decision letter https://doi.org/10.7554/eLife.84171.sa1
Author response https://doi.org/10.7554/eLife.84171.sa2

## Additional files

### Supplementary files

• Supplementary file 1. Histomorphometric analysis of WT *and Rspo3*$^{+/-}$ males.

• Supplementary file 2. Histomorphometric analysis of the tibia midshaft in 12 wk-old WT and Rspo3+/- females.

• Supplementary file 3. Histomorphometric analysis of the tibia midshaft in 12 wk-old WT and

*Rspo3*$^{+/-}$ males.

- Supplementary file 4. Histomorphometric analysis of 12 wk-old WT and Rspo3+/- vertebrae.
- Supplementary file 5. Histomorphometric analysis of 8 wk-old *Rspo3*$^{fl}$ and Rspo3-OB-cKO mice.
- Supplementary file 6. Histomorphometric analysis of 8 wk-old *Rspo3*$^{fl}$ and Rspo3-OB-cKO vertebrae.
- Supplementary file 7. Histomorphometric analysis of *Control*, *Rspo3*$^{+/-}$,*Dkk1-Tg* and *Rspo3*$^{+/-}$; *Dkk1-Tg* female mice at 6 wk of age.
- MDAR checklist

## Data availability

All data generated or analysed during this study are included in the manuscript and supporting file.

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
