## [Editor Report]

This seminal paper describes the divergent effects of Rspo3 haploinsufficiency on appendicular versus the axial skeletal in mice and, in doing so, highlights the differential regulation of the Wnt signaling pathway in a tissue-specific manner. Thus, while Rspo3 deficiency in osteoprogenitor cells increases bone mass in the appendicular skeleton, it causes osteopenia of vertebral bone. The study therefore not only identifies RSPO3, variants of which are associated with bone density in people, as a target for enhancing bone density in fracture-prone appendicular sites in the aging population, but also brings forth caution in the interpretation of single-site studies of the skeleton in most general terms.

---

## [Decision Letter]

**Decision letter after peer review:**

[Editors’ note: the authors submitted for reconsideration following the decision after peer review. What follows is the decision letter after the first round of review.]

Thank you for submitting the paper "R-spondin 3 deletion favors Erk phosphorylation to enhance Wnt signaling and bone formation" for consideration at *eLife*. Your submission has been assessed by a member of the Board of Reviewing Editors together with two reviewers. Although the work is of interest, we regret to inform you that the findings at this stage are considered too preliminary for further consideration at *eLife*.

Although both reviewers commended you on the quality of the data presented and the clarity of the presentation, they also had numerous concerns that had to do with what they perceived as a descriptive nature of the work, with discrepancy with other studies published on this topic, and a different interpretation of the DKK1-based experiments.

*Reviewer #1:*

R-spondins (Rspo1-4) are secreted proteins which have been characterized as positive regulators of the Wnt signaling in several cell types and model organisms. Wnt signaling, through its action on osteoblasts, plays an important role in the acquisition and maintenance of normal bone mass in vertebrates and GWAS studies have linked bone mineral density and fracture risk to the RSPO3 gene in humans. This study aims at determining the role of Rspo3 in osteoblasts using a series of mouse and cellular models. It was found that Rspo3 haplo-insufficiency (Rspo3+/-) in mice results in increase bone formation and bone mass, a result which disagree with the reported role of RSPO3 as a positive regulator of Wnt signaling. Inactivation of Rspo3 specifically in the osteoblast linage also results in a similar phenotype. A series of cell culture experiments shows that RSPO3 inactivation potentiates Wnt signaling possibly by increasing the activation of the ERK pathway.

Strengths of the study:

-Diverse approaches are used, including experiments in vivo (mouse models) and ex vivo taking advantage of various cell culture systems (osteoblasts, MEF).

-Both global and conditional mutants of Rspo3 are studied.

-Extensive bone histomorphometry is conducted to characterize the bone phenotypes of the mutant mice.

-The manuscript is well written.

Weaknesses of the study:

-The impact on bone of Rspo3 haploinsufficiency and osteoblast-specific inactivation presented here is in sharp contrast with a recent study by another group that detected a lower bone mass in mice lacking Rspo3 in osteoblasts (https://pubmed.ncbi.nlm.nih.gov/34389713/). The reason for this apparent discrepancy was not addressed or discussed in the manuscript.

-While histomorphometry is an accepted approach to measure bone density, the findings should be confirmed with another method, i.e., micro-CT. In addition, only one type of bone was examined (tibia) and the impact of Rspo3 deficiency on the axial skeleton was not addressed.

-Most of the mechanistic experiments involve Rspo3-/- MEF. To extend these findings to osteoblasts, these experiments should be repeated with Rspo3 fl/fl; Runx2-Cre BMSC cells.

-A number of important controls are missing. In particular, it will be important to show if recombinant mouse RSPO3 can rescue the cellular phenotypes of the Rspo3-/- cells.

Overall, without these critical missing pieces of data, the results presented do not fully support the claims of the manuscript.

Recommendations for the authors:

1) A recent paper by Nilsson et al. (https://pubmed.ncbi.nlm.nih.gov/34389713/), which is not cited in the current manuscript, shows that Rspo3 inactivation in osteoblasts with the same Runx2-Cre line, causes a low trabecular bone mass phenotype, at least in vertebra. These apparently contradictory results may be indicating that Rspo3 has different impact on bone depending on the skeletal sites. It will therefore be important to examine trabecular bone density in the vertebra in the current study to resolve this question. In addition, micro-CT should be used as an alternative approach to measure BV/TV in long bone and vertebra, since Nilsson et al. used this method.

2) Several experiments are comparing WT or Rspo3-/- MEF, but there is no evidence provided that WT MEF secrete significant amount of Rspo3. In addition, it will be essential to show that the cellular phenotypes of the Rspo3-/- BMSC and MEF presented in Figure 4-7 can be rescued by the exogenous addition of recombinant mouse Rspo3. Otherwise, it is not possible to conclude that the changes in Wnt signaling detected in these cells is really caused by a cell-autonomous effect of the absence of Rspo3. In link with this comment, there are inconsistency between some of the data presented. For example, in figure 5A, exogenous Wnt3a is required to induce Tcf1 promoter, measured using TOPFlash luciferase assay, and the addition of recombinant Rspo3 do not suppressed the induction. In contrast, in figure 5B, Rspo3-/- MEF have increased Tcf1 expression in absence of Wnt3a treatment.

3) Linked to the previous comment, cell culture experiments (Figure 4-7) should be repeated with fl/fl;Runx2-Cre BMSC or with Rspo3 fl/fl calvaria osteoblasts transduced with Cre ex vivo.

4) If the global haploinsufficiency of Rspo3 results in an increased bone mass through a cell-autonomous effect on osteoblasts, then it is expected that an osteoblast specific haploinsufficiency (Rspo3 +/flox ; Runx2-Cre) will also results in a high bone mass phenotype. Yet, this genotype was not analyzed.

4) Whether global Rspo3 haploinsufficiency results in decreased expression of Rspo3 in osteoblasts and in bone was not shown.

5) An important control to be included in Figure 3 is the +/+ ; Runx2-Cre mice to exclude any potential non-specific effect of this Cre line on bone mass.

6) A description of how the Rspo3 floxed mice used in figure 3 were generated is not included in the material and methods and no reference is provided in link with this line. How this model compares to the one published by Nilsson et al. (see point 1)?

7) The rational for using recombinant human Wnt3a, Dkk1 and Rspo3 on mouse osteoblast and MEF cultures is not clear, since the equivalent mouse proteins are available from R&D System.

8) The interpretation of the data presented in Figure 6.D-E (Rspo3+/-;TgDkk1) is challenging. Since these two mouse lines have opposite phenotypes in term of bone mass, the normalization of the bone density in the double transgenic does not necessarily provide a clear demonstration that the two genes act in the same pathway. Moreover, the TgDkk1 overexpress Dkk1 in osteoblasts (col1a1 promoter) while the haploinsufficiency of Rspo3 is in all cells, further complicating the interpretation of this genetic experiment.

9) In Figure 6A there is only n=2 per group. Although one can run ANOVA or t test on n=2, it will be more appropriate to have additional replicates.

*Reviewer #2:*

The work entitled "R-spondin 3 deletion favors Erk phosphorylation to enhance Wnt signaling and bone formation" by Nagano et al. investigated the role of RSPO3 in skeletal homeostasis. The authors analyzed bone formation and bone mass in RSPO3 haplo-insufficient mice and in mice lacking RSPO3 in osteoblasts. They found an increased bone mass. They propose that this phenotype is driven by an increase WNT and ERK signaling in osteoblast progenitor cells. The in vivo data clearly demonstrate that Rspo3 controls bone formation and high bone mass. However, in its current status, the work is descriptive and lacks the levels of novelty and mechanistic insights that are expected by readers of e*Life*. Indeed the regulation of both DKK/WNT and ERK signaling by RSPO3 has been previously described. The authors confirmed these observations without providing additional mechanistic insights into this regulation. As result the work appears quite descriptive/preliminary. Importantly, the in vivo data showing that Rspo3 haplo-insufficiency counteracted the effect of Dkk1 overexpression in vivo, clearly indicate that Dkk1 is not involved in RSPO3 phenotype, since Dkk1 overexpression does not have any effect on RSPO3 high bone mass phenotype. Thus most likely, Dkk1 does not play significant roles in RSPO3-mediated regulation of bone mass.

In addition, the link between ERK and DKK1 signaling regulation by RSPO3 has not been explored, so it is unclear whether these two pathways act in parallel or not.

The authors propose that the high bone mass phenotype observed in Rspo3 haplo-insufficient mice wes due to an increase in bone marrow precursor cells (mesenchymal stromal cells (MSC) population (Lin-CD45-CD31-CD51+Sca-1+)). This phenotype was mirrored by the RUNX2-Cre mediated deletion of Rspo3. Do MSC cells express Runx2-Cre? This should be verified by the authors.

---

## [Author Response]

[Editors’ note: the authors resubmitted a revised version of the paper for consideration. What follows is the authors’ response to the first round of review.]

Although both reviewers commended you on the quality of the data presented and the clarity of the presentation, they also had numerous concerns that had to do with what they perceived as a descriptive nature of the work, with discrepancy with other studies published on this topic, and a different interpretation of the DKK1-based experiments.Reviewer #1:R-spondins (Rspo1-4) are secreted proteins which have been characterized as positive regulators of the Wnt signaling in several cell types and model organisms. Wnt signaling, through its action on osteoblasts, plays an important role in the acquisition and maintenance of normal bone mass in vertebrates and GWAS studies have linked bone mineral density and fracture risk to the RSPO3 gene in humans. This study aims at determining the role of Rspo3 in osteoblasts using a series of mouse and cellular models. It was found that Rspo3 haplo-insufficiency (Rspo3+/-) in mice results in increase bone formation and bone mass, a result which disagree with the reported role of RSPO3 as a positive regulator of Wnt signaling. Inactivation of Rspo3 specifically in the osteoblast linage also results in a similar phenotype. A series of cell culture experiments shows that RSPO3 inactivation potentiates Wnt signaling possibly by increasing the activation of the ERK pathway.Strengths of the study:-Diverse approaches are used, including experiments in vivo (mouse models) and ex vivo taking advantage of various cell culture systems (osteoblasts, MEF).-Both global and conditional mutants of Rspo3 are studied.-Extensive bone histomorphometry is conducted to characterize the bone phenotypes of the mutant mice.-The manuscript is well written.Weaknesses of the study:-The impact on bone of Rspo3 haploinsufficiency and osteoblast-specific inactivation presented here is in sharp contrast with a recent study by another group that detected a lower bone mass in mice lacking Rspo3 in osteoblasts (https://pubmed.ncbi.nlm.nih.gov/34389713/). The reason for this apparent discrepancy was not addressed or discussed in the manuscript.-While histomorphometry is an accepted approach to measure bone density, the findings should be confirmed with another method, i.e., micro-CT. In addition, only one type of bone was examined (tibia) and the impact of Rspo3 deficiency on the axial skeleton was not addressed.

Thank you for your comment and for calling our attention on the manuscript by Nilsson et al. Note however that this paper was published while our manuscript was in review, explaining that we did not mention or discuss their findings. It is however important to note that the Ohlsson group reported only axial skeleton (vertebrae) data and at only one time point (12 week), when we had reported only appendicular skeleton (tibia) data, which showed consistent increase in bone mass at least 3 different ages.

Thus, in response to your comments and Ohlsson group’s paper, we now have analyzed the vertebrae in the same animals for which we reported tibia data. This led to the very surprising finding that if in long bones the Runx2-cre driven deletion of *Rspo3* led to high bone mass, the vertebrae of the same animals showed a lower bone mass, in agreement with the findings reported by Nilsson *et al.* Thus, looking at the same part of the skeleton as Nilsson *et al.,* we find the same results. Importantly, Nilsson *et al.* did not examine the appendicular skeleton and our data, confirmed in several in vivo models and in vitro studies, confirms that *Rspo3* haploinsufficiency and its deletion in osteoblast progenitor can also lead to increased bone formation and bone mass. Notably also, examination of the vertebrae in *Rspo3^+/-^* mice, where we found high bone mass in tibiae, showed no significant changes, despite a trend to increased bone formation. Thus, the effects of *Rspo3* global deletion or its targeted deletion in osteoprogenitors differ between the axial and the appendicular skeleton.

It is not easy to explain such a surprising finding, and few papers in the literature examined both sites in animal studies. We have now added this data and discussed various possibilities that could explain the differences: crosstalk with load bearing (vertebrae are less loaded than limbs), the differential expression levels of receptors and co-receptors, distinct embryonic origin as well as of agonists and antagonists and the presence of distinct populations of stem cells and non-stem progenitors in the vertebrae and in the long bones, are examples of potential explanations. Yet, these findings do not invalidate our very strong and abundant set of data showing that, at least in vitro and in vivo in long bones, Rspo3 haploinsufficiency and OB-targeted deletion can also lead to high bone formation and bone mass.

On your other point, we had previously collected microCT data and have now increased this data to include both long bones and vertebrae.

-Most of the mechanistic experiments involve Rspo3-/- MEF. To extend these findings to osteoblasts, these experiments should be repeated with Rspo3 fl/fl; Runx2-Cre BMSC cells.

We show that BMSCs isolated from *Rspo3* haploinsufficient mice have increased canonical Wnt signaling (new Figure 5) and increased CFU-F and CFU-OB potential (Figure 2). We believe that this set of experiments together with our in vivo data in *Rspo3^+/-^* mice and targeted deletion in Runx2^+^ cells, establish a cell autonomous role of Rspo3. We do not think that to repeat these studies in the Runx2-Cre model would add much.

-A number of important controls are missing. In particular, it will be important to show if recombinant mouse RSPO3 can rescue the cellular phenotypes of the Rspo3-/- cells.Overall, without these critical missing pieces of data, the results presented do not fully support the claims of the manuscript.

Thank you for this useful suggestion. We have now performed such rescue experiments and show now (new Figure 2) that recombinant Rspo3 prevents (rescues) the increase in CFU-OB number seen with Rspo3 haploinsufficiency, confirming that Rspo3 can also have a negative impact on OB differentiation.

Recommendations for the authors:1) A recent paper by Nilsson et al. (https://pubmed.ncbi.nlm.nih.gov/34389713/), which is not cited in the current manuscript, shows that Rspo3 inactivation in osteoblasts with the same Runx2-Cre line, causes a low trabecular bone mass phenotype, at least in vertebra. These apparently contradictory results may be indicating that Rspo3 has different impact on bone depending on the skeletal sites. It will therefore be important to examine trabecular bone density in the vertebra in the current study to resolve this question. In addition, micro-CT should be used as an alternative approach to measure BV/TV in long bone and vertebra, since Nilsson et al. used this method.

These points have now been addressed (see above).

2) Several experiments are comparing WT or Rspo3-/- MEF, but there is no evidence provided that WT MEF secrete significant amount of Rspo3. In addition, it will be essential to show that the cellular phenotypes of the Rspo3-/- BMSC and MEF presented in Figure 4-7 can be rescued by the exogenous addition of recombinant mouse Rspo3. Otherwise, it is not possible to conclude that the changes in Wnt signaling detected in these cells is really caused by a cell-autonomous effect of the absence of Rspo3. In link with this comment, there are inconsistency between some of the data presented. For example, in figure 5A, exogenous Wnt3a is required to induce Tcf1 promoter, measured using TOPFlash luciferase assay, and the addition of recombinant Rspo3 do not suppressed the induction. In contrast, in figure 5B, Rspo3-/- MEF have increased Tcf1 expression in absence of Wnt3a treatment.

Levels of Rspo3 in MEF are presented in Figure 6-supplement 1b and by RT-QPCR cycle number for Rspo3 in MEF is between 22 and 24. We show now that recombinant Rspo3 prevents (rescues) the increase in CFU-OB number seen with Rspo3 haploinsufficiency (new Figure 2).

The reviewer is correct in mentioning that in the old Figure 5a (revised Figure 6-supplement 1b), Wnt3a induces the Tcf1 promoter and that Rspo3 functions as a potentiator of Wnt3a activity in wt MEFs. It is important to note however that Rspo3 by itself does not lead to Wnt signaling activation in the absence of Wnt3a. Here, our studies in BMSCs, in long bones and in MEFs, shown in the new Figure 5, 6 and 8, clearly and unexpectedly demonstrate that absence of *Rspo3* results by itself in a significant activation in Wnt signaling at steady state, as showed by an increase in Tcf1 and Axin2 expression levels and active β-catenin and pLrp6 protein levels. We are aware that this is the opposite of what one could expect when knocking down or deleting *Rspo3*, but this is what makes it the key point of this paper. Several of our independent in vivo and in vitro experiments demonstrate that this is indeed the case, making this paper most novel and interesting. This is detailed in our discussion.

3) Linked to the previous comment, cell culture experiments (Figure 4-7) should be repeated with fl/fl;Runx2-Cre BMSC or with Rspo3 fl/fl calvaria osteoblasts transduced with Cre ex vivo.

These points have now been addressed (see above).

4) If the global haploinsufficiency of Rspo3 results in an increased bone mass through a cell-autonomous effect on osteoblasts, then it is expected that an osteoblast specific haploinsufficiency (Rspo3 +/flox ; Runx2-Cre) will also results in a high bone mass phenotype. Yet, this genotype was not analyzed.

Please note that in the long bones Runx2creRspo^fl^ induces only a 50% decrease in *Rspo3* gene expression, in fact mimicking a haplo-insufficiency.

4) Whether global Rspo3 haploinsufficiency results in decreased expression of Rspo3 in osteoblasts and in bone was not shown.

Thank you for your comment. These control data have now been added in Figure 1a and in Supplemental Figure 1-supplement 1a.

5) An important control to be included in Figure 3 is the +/+ ; Runx2-Cre mice to exclude any potential non-specific effect of this Cre line on bone mass.

As shown in the new Figure 3 and 4 – Supplement 1, microCT analysis shows no significant differences in BV/TV, Tb.N, Th.th. and Tb.S in both L5 and Femur between *Runx2Cre, wt* and *Rspo3^fl^* mice.

6) A description of how the Rspo3 floxed mice used in figure 3 were generated is not included in the material and methods and no reference is provided in link with this line. How this model compares to the one published by Nilsson et al. (see point 1)?

A description of the strategy used to generate the Rspo3 mutant mice is reported in reference#27 (Kazanskaya O, Ohkawara B, Heroult M, Wu W, Maltry N, Augustin HG, Niehrs C. The Wnt signaling regulator R-spondin 3 promotes angioblast and vascular development. Development) Reference 27 was and is mentioned in the methods in the Animal section.

Floxed mice and global het mice were generated using the same targeting strategy.

Briefly, as detailed in Suppl. Figure S2 in reference # 27, the Rspo3 mutant mice used in our studies were generated by flanking exon1 with loxP sequences. The fragment flanked by the 2 loxP genes was then removed by cre-recombinase to obtain a Rspo3 haploinsufficient mouse colony used to generate flox mice for targeted deletion. The Rspo3 flox mice used by Nilsson *et al.* were generated by flanking exons 2-4 with Loxp sequences and recombined by Cre recombinase.

7) The rational for using recombinant human Wnt3a, Dkk1 and Rspo3 on mouse osteoblast and MEF cultures is not clear, since the equivalent mouse proteins are available from R&D System.

In response to this comment, there was not specific rationale to use human or mouse recombinant proteins. Both have been shown to work in mouse cells.

8) The interpretation of the data presented in Figure 6.D-E (Rspo3+/-;TgDkk1) is challenging. Since these two mouse lines have opposite phenotypes in term of bone mass, the normalization of the bone density in the double transgenic does not necessarily provide a clear demonstration that the two genes act in the same pathway. Moreover, the TgDkk1 overexpress Dkk1 in osteoblasts (col1a1 promoter) while the haploinsufficiency of Rspo3 is in all cells, further complicating the interpretation of this genetic experiment.

Thank you for this comment. We agree with the reviewer that interpretation of such genetic experiments can be complex. Nonetheless our data show clearly that Rspo3 haploinsufficiency can rescue the low bone mass phenotype of mice overexpressing Dkk1 in osteoblasts, indicating that Rspo3 deletion effects occur in the osteoblast micro-environment, findings strongly supported by our in vitro data showing that when Rspo3 is deleted the efficiency of Dkk1 in blocking Wnt signaling activation is decreased.

9) In Figure 6A there is only n=2 per group. Although one can run ANOVA or t test on n=2, it will be more appropriate to have additional replicates.

As requested, we have added 1 extra experiment and we have now n=3 (now in the new Figure 7a). Note that in Figure 7a, the statistical analysis performed is Student T-test. This is reflected in the legend.

Reviewer #2:The work entitled "R-spondin 3 deletion favors Erk phosphorylation to enhance Wnt signaling and bone formation" by Nagano et al. investigated the role of RSPO3 in skeletal homeostasis. The authors analyzed bone formation and bone mass in RSPO3 haplo-insufficient mice and in mice lacking RSPO3 in osteoblasts. They found an increased bone mass. They propose that this phenotype is driven by an increase WNT and ERK signaling in osteoblast progenitor cells. The in vivo data clearly demonstrate that Rspo3 controls bone formation and high bone mass. However, in its current status, the work is descriptive and lacks the levels of novelty and mechanistic insights that are expected by readers of eLife. Indeed the regulation of both DKK/WNT and ERK signaling by RSPO3 has been previously described. The authors confirmed these observations without providing additional mechanistic insights into this regulation. As result the work appears quite descriptive/preliminary. In addition, the link between ERK and DKK1 signaling regulation by RSPO3 has not been explored, so it is unclear whether these two pathways act in parallel or not.

The main novelty of our report is that deletion of *Rspo3* would be expected, based on current knowledge, to decrease bone formation and bone mass by removing a Wnt co-activator. In contrast, we demonstrate instead that decreased levels of Rspo3 do increase bone formation and bone mass, and this is in several independent in vivo genetic and in vitro models. Moreover, to our knowledge there are no previous data showing that in vivo haploinsufficiency of *Rspo3* can rescue the low bone mass seen in mice with osteoblast targeted overexpression of Dkk1.

Our signaling findings also add to the current models. As mentioned in the manuscript, previous reports showed that Rspo3 deletion in vitro (siRNA) in adipose precursor cells leads to increased pERK phosphorylation and increased cell differentiation. We confim here activation of pERK by low Rspo3 levels but the novelty of our studies lies in that we report a direct link to Wnt signaling, establishing a novel Rspo3/ERK/Wnt axis.

We show that in the absence of Rspo3 the increase in pERK phosphorylation contributes to Wnt signaling activation as indicated by the findings that blocking ERK signaling significantly impairs Wnt signaling activation seen in absence of Rspo3. Therefore, our studies add Wnt to the Rspo3/ERK axis.

Importantly, the in vivo data showing that Rspo3 haplo-insufficiency counteracted the effect of Dkk1 overexpression in vivo, clearly indicate that Dkk1 is not involved in RSPO3 phenotype, since Dkk1 overexpression does not have any effect on RSPO3 high bone mass phenotype.Thus most likely, Dkk1 does not play significant roles in RSPO3-mediated regulation of bone mass.

We respectfully disagree with this comment. Our data show (1) haploinsufficiency of *Rspo3* rescues significantly the low bone mass phenotype seen in *TgDkk1* mice and (2) However, in contrast to what is stated by the reviewer (“Dkk1 overexpression does not have any effect on RSPO3 high bone mass phenotype”) overexpression of Dkk1 in osteoblasts decreased significantly the high bone mass phenotype seen in *Rspo3* het mice, validating our interpretation of the results.

The authors propose that the high bone mass phenotype observed in Rspo3 haplo-insufficient mice wes due to an increase in bone marrow precursor cells (mesenchymal stromal cells (MSC) population (Lin-CD45-CD31-CD51+Sca-1+)). This phenotype was mirrored by the RUNX2-Cre mediated deletion of Rspo3. Do MSC cells express Runx2-Cre? This should be verified by the authors.

The reviewer is correct: we show that global haploinsufficiency of Rspo3 leads to an increase in BM osteoblast precursors and as shown by bone histomorphometry, to an increase in the number of osteoblasts in Rspo3 het mice. The population of MSC in Runx2creRspo3^fl^ mice was not investigated by flow cytometry. However, deletion of Rspo3 in Runx2^+^ cells also led to a significant increase in the number of osteoblasts by bone histomorphometry.